



# A computationally efficient engineering aerodynamic model for swept wind turbine blades

Ang Li[1], Georg Raimund Pirrung[1], Mac Gaunaa[1], Helge Aagaard Madsen[1], and Sergio González Horcas[1]

[1]Department of Wind Energy, Technical University of Denmark, Frederiksborgvej 399, 4000 Roskilde, Denmark

**Correspondence:** Ang Li (angl@dtu.dk)

**Abstract.** In this work, a computationally efficient engineering model for the aerodynamics of swept wind turbine blades is proposed for the extended blade element momentum (BEM) formulation. The model is modified based on a coupled near- and far-wake model, in which the near-wake is assumed to be the first quarter revolution of the non-expanding helical wake of the own blade. For the special case of in-plane trailed vorticity, the original empirical equations determining the steady-state value of the near-wake induction are replaced by the analytical results, which are in the form of incomplete elliptic integrals. For the general condition of helical trailed vorticities, the steady-state near-wake induction is approximated based on the results of the special conditions and a correction factor. The factor is calculated using empirical equations with influence coefficient tensors, to minimize the computational efforts. These influence coefficient tensors are pre-calculated and are fitted to the results from the numerical integration of the Biot-Savart law. With the indicial function approach, it is not necessary to explicitly save the information of the vorticities that were trailed in the previous time steps. This engineering approach is a combination of analytical results and numerical approximations, with low and constant computational effort for each time step. The proposed model is practically applicable to time-marching aero-servo-elastic simulations. The results of the swept blades with uniform inflow perpendicular to the rotor calculated from the proposed model are compared with the results from a BEM code, a lifting-line solver as well as a Navier-Stokes solver. The significantly improved agreement with the higher-fidelity models compared to the BEM method highlights the performance of the proposed method.

## 1 Introduction

With the technological advancements in the design optimization and manufacturing of horizontal-axis wind turbines, the turbine blades are becoming increasingly flexible. Thus, there could be significant in-plane and out-of-plane deformations due to the aeroelastic loads. In addition, there is an increasing interest in the backward swept blades because of the possibility to achieve passive load alleviation with geometric bend-twist coupling (Liebst, 1986; Zuteck, 2002; Larwood and Zutek, 2006; Larwood et al., 2014; Manolas et al., 2018). The recent research by Barlas et al. (2021) is on the aeroelastic design optimization of





blade tip add-ons with curved shapes. Higher-fidelity tools such as lifting-line solvers (LL) and fully-resolved Navier-Stokes solvers (often referred to as Computational Fluid Dynamics, CFD), are limited in the application of design optimization and in repetitive aeroelastic load calculations, due to their high computational cost.

In the spectrum of the lower-fidelity models, the most commonly used blade element momentum (BEM) method implicitly
assumes a planar rotor with straight blades. If the actuator disc (AD) is not planar, the induction deviates from what the BEM model predicts as demonstrated by Madsen and Rasmussen (1999) using a CFD model for computation of the AD flow. Further, the disc approach in the BEM method has some fundamental shortcomings in the capability to model response to turbulent inflow, stability and steep load variations along the blade like partial pitch actuation.

This led to the formulation of the coupled model (usually referred to as the near-wake model) by Madsen and Rasmussen
(2004), which is a hybrid model of a lifting-line method and the BEM method. It combines the detailed modelling of the local blade aerodynamics in the lifting-line model using a simplified approach and the far-wake modelling by the BEM method. The near-wake is defined to be the first quarter revolution of the trailed vorticity of the own blade, which is modelled as non-expanding helical vortex filaments. The near-wake induction is approximated using empirical equations and correction factors. The indicial function approach is used, so that the information of the vorticities trailed from the previous time steps are
not explicitly stored. Then, the computational effort is relatively low and is independent of the elapsed simulation time. The remaining trailed vorticity of the wind turbine vortex system is defined as the far-wake and is modelled by a far-wake BEM model (Madsen and Rasmussen, 2004). The near-wake model and the far-wake model are coupled together with a coupling factor to get the total induction (Andersen et al., 2010; Pirrung et al., 2016).

Since the first version of the model in 2004, there have been several improvements. Integration in the multibody aeroelastic
HAWC2 code is presented in Andersen et al. (2010) and further developments of the model are presented in Pirrung et al. (2014), Pirrung et al. (2016) and Pirrung et al. (2017a). However, the model in its latest version (Pirrung et al., 2017a) still assumes straight blades and is not able to correctly model the aerodynamics of the swept blades. This is the further development of the model to be presented in the present work.

There has been previous work by Li et al. (2018) on this topic, in which the good performance of the modified coupled
model on the prediction of the aerodynamic loads of the swept blades is demonstrated. However, in that work, the near-wake induction is calculated by directly integrating the Biot-Savart law numerically. This approach is computationally expensive and is not suitable for the application to aeroelastic simulations. In addition, the method of modelling the curved bound vorticity influence on itself in that previous work was incomplete and limited to swept blades. The updated method of modelling the influence of curved bound vortex is described in detail later by Li et al. (2020).

In the present work, the background of the engineering aerodynamic models for horizontal-axis wind turbines are first briefly described. Then, the details of the near-wake model, including the analytical solutions as well as the engineering approaches for a computationally efficient implementation, are described. Afterwards, the far-wake model and the coupling method are briefly discussed. Finally, the aerodynamic loads of the swept blades under the special condition of uniform inflow perpendicular to the rotor plane predicted by the proposed model are compared with the results from a BEM code, a lifting-line solver as well
as a CFD Reynolds-averaged Navier–Stokes (RANS) solver.



## 2 Background: Engineering aerodynamic models

For the application of aeroelastic simulations of wind turbines, there are multiple low- and mid-fidelity engineering aerodynamic models with different assumptions. An example of a low-fidelity model is the polar grid implementation of the blade element momentum (BEM) method with unsteady aerodynamics (Madsen et al., 2020). For the computation of the induction,

the momentum part of BEM, the swept area is assumed to be a planar surface and form an actuator disc (AD). However, all computations of the aerodynamic forces in the blade element part of BEM, as input to induction computations, are carried out for the actual blade shapes taking into account in-plane sweep and out-of-plane shape. The momentum theory and the angular momentum theory are applied to balance the out-of-plane loads as well as the in-plane loads between the AD and the flow. The evaluation of induction is carried out at each time step on a stationary polar grid covering the AD (Madsen et al., 2020).

When the blade has no prebend and it is straight, the version of the BEM method that excludes the drag force in the momentum balancing is equivalent to a vortex cylinder model that excludes the wake rotation effect (Branlard and Gaunaa, 2015). It is also argued by Branlard (2017) that the proper way of implementing the BEM method should exclude the drag force during the momentum balancing from which the induced velocities are determined. This means in the BEM method, the wake of the rotor is equivalently modelled with non-expanding concentric vortex cylinders. This also implicitly shows that the BEM method

assumes the blades are straight and stay in the rotor plane.

An example of the higher-fidelity model is the lifting-line method, which models each blade of the rotor with a bound vortex line. This is under the assumption that the bound vorticity of a blade is concentrated into a line vortex at the quarter-chord line. Vortices are trailed from the bound vortex line, with the trailed vorticity strength equal to the spanwise gradient of the bound vorticity. The trailed vortices are modelled with helical vortex filaments and could possibly include the wake expansion effect.

There is also shed vorticity for the unsteady conditions. Comparing to the BEM method, the lifting-line method models the blade and the wake using vortex line filament and helical vortex filaments instead of using superposition of actuator discs and concentric vortex cylinders. The assumption that the blades are straight and are located in the rotor plane can be relaxed. In addition, the influence of the non-straight bound vortex on itself should also be explicitly included (Li et al., 2020). With this bound vorticity correction, the lifting-line method is able to correctly model the influence of the blade sweep and dihedral.

The coupled near- and far-wake model is considered as a hybrid of the aforementioned two methods. For the first quarter revolution of the own wake of every blade, which corresponds to the near-wake, the model is similar to the lifting-line method without wake expansion. In the modified coupled model by Pirrung et al. (2016), the bound vortex line located at the quarter-chord line is assumed to be straight and stays in the rotor plane. The trailed vorticity emanates from it and forms the non-expanding helical wake with the rotation of the blade. The remaining wake, including the own wake of the blade after the first

quarter revolution and also the wake of other blades, is defined as the far-wake. The far-wake is modelled by a far-wake BEM model (Madsen and Rasmussen, 2004) that does not account for Prandtl's tip-loss correction. The idea is similar to using the vortex cylinder model as the far-wake model, in which the vortex cylinders begin further downstream compared to the rotor plane. The near-wake model and the far-wake model are coupled together with a coupling factor (Andersen et al., 2010; Pirrung et al., 2016). The coupling factor is computed so that the aerodynamic thrust of the whole rotor calculated from the coupled



model is comparable to that calculated from the BEM method. The different ideas of modelling the blades and the wake in the three different engineering aerodynamic models are illustrated in Fig. 1.

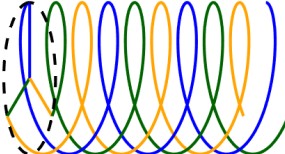 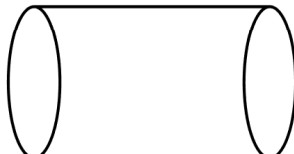 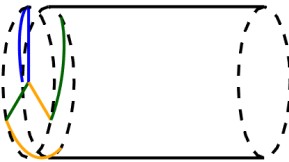

**Figure 1.** Illustration of the modelling of the blade and the wake in the three different engineering aerodynamic models. In the lifting-line method (left), each blade is modelled by a bound vortex line, and the trailed vorticity is modelled with helical vortex filaments. In the generalized actuator disc model, such as the BEM method (middle), the blades are modelled by superposition of actuator discs with the aid of a tip correction model. The vortices are trailed from the rotor plane and form concentric cylindrical vortex wakes. Only the tip vortex is shown in the figure. In the coupled near- and far-wake model (right), the blades and the near-wake are modelled similar to the lifting-line method while the far-wake is modelled similar to the BEM method.

In the modified coupled model proposed in the present work, the assumption of straight blades in the original coupled model is partially relaxed. The bound vortex can be curved but is constrained to the rotor plane, which means the blades can be swept
forward or backwards. There are two key features of the modified model and they correspond to two impacts of the blade sweep on the vortex system. The first one is the influence of the curved bound vortex on itself, which has been described by Li et al. (2020). It has been shown that the influence of the curved bound vortex should be explicitly modelled for the generalized lifting-line methods that use 2-D airfoil data. The influence is modelled by including the difference of the 3-D induction of the curved bound vortex and the 2-D induction evaluated at the three-quarter-chord point. The method is applicable to both
the modified coupled near- and far-wake model and the lifting-line method. The second feature is the in-plane-shifted starting position of the trailed vorticity due to the blade sweep, which will be discussed in detail in this work. The calculation points and the trailing points are located on the curved bound vortex line, which is following the quarter-chord line of the swept blade. The trailed vorticities emanate from the trailing points and will then be shifted forward or backwards compared to the calculation points due to the non-straight bound vortex. The relatively shifted position of the trailed vorticity compared to the
straight blade will change the steady-state near-wake induction.

The modified near-wake model is similar to the modified lifting-line model for curved wind turbine blades that is labelled as LL-test in Li et al. (2020). The calculation points for the trailed vorticity induction are placed on the quarter-chord line, which is also the (curved) bound vortex line. This can be justified by the comparison of the results of swept blades from different versions of the lifting-line methods with the Navier-Stokes solver, as performed in Li et al. (2020). If the curved bound vortex
influence is explicitly modelled, the results from the lifting-line methods are in good agreement with the higher-fidelity Navier-Stokes solver. This is true irrespectively of the location of point used for the calculation of the trailed vorticity induction (i.e. quarter-chord or three-quarter chord).

## 3 Trailing function

The trailing function represents the induction due to an elementary trailed vorticity arc, depending on its azimuthal location relative to the blade. In a previous work (Li et al., 2018), the trailing functions of the axial and the tangential induction of a counter-clockwise rotating swept blade have been derived using the Biot-Savart law. In this section, the trailing functions for a clockwise rotating swept blade, whose rotational vector is in the downwind direction, are derived. The coordinate system as well as the geometry of the trailed vortex are clarified in a consistent manner. In addition, the steady-state near-wake induction is also defined and the analytical expressions for some special cases are derived in Appendix B1 and B2.

The coordinate system used in the present work is consistent with the commonly used conventions for wind turbine aerodynamics. In this work, we assume the blade has no prebend, which means the out-of-plane component of the geometry is assumed to be zero. However, if prebend exists, the projection of the blade main-axis into the rotor plane should be used to calculate the sweep geometry for the input of the model proposed here. The origin of the coordinate system is located at the rotational center of the rotor, and it is locally defined for every blade and every section. The $z$-axis is defined from the rotational center to the calculation point of any given section. The $x$-axis is common for every blade and section. It is parallel to the rotor axis, and it is positive in the upwind direction. The $y$-axis is normal to both $x$-axis and $z$-axis, and its direction is defined so that a right-handed system is found. For different sections, the corresponding coordinate systems are rotated about the $x$-axis, so that the calculation point $s$ is always located on the $z$-axis. In Fig. 2, the front view of a clockwise rotating backward swept blade and its trailed helical vortex are shown to illustrate the coordinate system and the geometric variables.

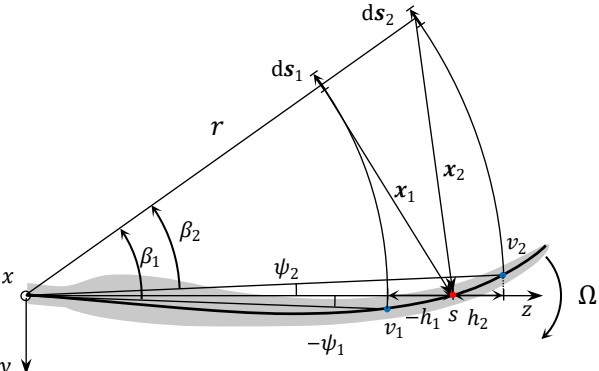

**Figure 2.** The front view of a backward swept wind turbine blade that is rotating clockwise with the rotational speed $\Omega$ and its trailed vortex. The calculation point $s$ is located on the $z-$axis. Two conditions are shown for this backward swept blade. Firstly, when the trailing point $v_1$ is further inboard compared to the calculation point, the relative distance $h_1$ and the sweep angle $\psi_1$ are both smaller than zero. Secondly, when the trailing point $v_2$ is further outboard, the relative distance $h_2$ and the sweep angle $\psi_2$ are both larger than zero. The difference of the azimuthal angle of the elementary trailed vorticity $\mathrm{d}s$ with respect to the trailing point is $\beta$. The position vector $x$ is pointing from the elementary trailed vortex $\mathrm{d}s$ to the calculation point $s$. Please note, the position vector $x$ and the elementary trailed vorticity $\mathrm{d}s$ shown in the sketch are their projection into the rotor plane.



The radius of the trailing point is noted as $r$. The radius of the calculation point is $r_{cp}$. The difference between the radius of the calculation point and the trailing point, which is equal to $r$ minus $r_{cp}$, is noted as $h$. The sweep angle is $\psi$, which is defined as the difference between the azimuthal angle of the calculation point and the trailing point. The elementary trailed vortex filament $\mathrm{d}\boldsymbol{s}$ is positive for trailed vorticity with positive strength when pointing away from the blade since the blade is

rotating clockwise. The position vector $\boldsymbol{x}$ is pointing from the elementary trailed vortex filament $\mathrm{d}\boldsymbol{s}$ to the calculation point $s$. The azimuthal difference of the elementary trailed vorticity with respect to the trailing point is $\beta$, which corresponds to the azimuthal angle that the elementary trailed vorticity has travelled. The rotational speed of the blade is $\Omega$.

It is assumed that the near-wake part of the trailed vorticity convects downstream with the velocity determined at the blade. This is because the first quarter revolution of the wake is generally very close to the rotor plane where it is emitted. The in-plane

and out-of-plane components of the flow velocity at the trailing points are $v_{ip}$ and $v_{oop}$, respectively:

$$v_{ip} = \Omega r + v_{ip}^{motion} + v_{ip}^{ind} \tag{1}$$

$$v_{oop} = U_\infty + v_{oop}^{motion} + v_{oop}^{ind} \tag{2}$$

where the relative flow velocities from the induction and the blade motion are included in the velocity. They are noted with the superscripts *ind* and *motion*, respectively.

The $z-$component is the radial component of the velocity and is not considered in this study. This is because for the swept blades, the radial velocity contributes to the large in-plane component of the relative velocity seen by the 2-D airfoil section. The contribution is also linearly proportional to the sine of the sweep angle. Since for ordinary operation conditions, the flow angle is small, the influence of the radial velocity on the flow angle and consequently on the lift and drag force of the 2-D section is negligible.

The relative velocity $V_{rel}^{tp}$ and the helix angle $\varphi$ of the trailed vorticity are determined by the velocity vector at the trailing point on the blade.

$$V_{rel}^{tp} = \sqrt{v_{oop}^2 + v_{ip}^2} \tag{3}$$

$$\varphi = \arctan\left(\frac{v_{oop}}{v_{ip}}\right) \tag{4}$$

In the previous work by Pirrung et al. (2016) and later by Li et al. (2018), the tangential speed $\Omega r$ due to rotation is used as

$v_{ip}$. The in-plane induced velocity is generally much smaller than the tangential speed $\Omega r$. When the in-plane motion $v_{ip}^{motion}$ is small compared to $\Omega r$, the results using either the full value for the in-plane velocity or only $\Omega r$ will be similar.

Assuming both $v_{ip}$ and $v_{oop}$ are constant, the elapsed time $\Delta t$ resulting from the trailed vorticity element $\mathrm{d}\boldsymbol{s}$ traveling an azimuthal angle of $\beta$ is:

$$\Delta t = \frac{r\beta}{v_{ip}} \tag{5}$$



The $x-$component of the position vector $\boldsymbol{x}$ that is pointing from the elementary trailed vorticity $\mathrm{d}\boldsymbol{s}$ to the calculation point $s$ is:

$$x_x = v_{oop}\Delta t = \frac{v_{oop}}{v_{ip}}r\beta \tag{6}$$

The other components of the two vectors of $\boldsymbol{x}$ and $\mathrm{d}\boldsymbol{s}$ that are used to determine the induction function, can be easily

determined. They are expressed as function of $h$, $r$, $\psi$, $\beta$ and $\varphi$ as follows:

$$\boldsymbol{x} = \begin{pmatrix} r\beta\tan\varphi \\ r\sin\left(\beta+\psi\right) \\ r-h-r\cos\left(\beta+\psi\right) \end{pmatrix} \tag{7}$$

$$\mathrm{d}\boldsymbol{s} = \mathrm{d}s\cos\varphi \begin{pmatrix} -\tan\varphi \\ -\cos\left(\beta+\psi\right) \\ -\sin\left(\beta+\psi\right) \end{pmatrix} \tag{8}$$

For the infinitesimally trailed vorticity element $\mathrm{d}\boldsymbol{s}$ with strength $\Delta\Gamma$, the induced velocity at the blade section $s$ due to this trailed vortex element is calculated according to the 3-D Biot-Savart law. The minus sign in the equation is due to the definition

of $\boldsymbol{x}$ that is pointing from the elementary trailed vorticity to the calculation point.

$$\mathrm{d}\boldsymbol{w} = -\frac{\Delta\Gamma}{4\pi}\frac{\boldsymbol{x}\times\mathrm{d}\boldsymbol{s}}{\|\boldsymbol{x}\|^3} \tag{9}$$

The elementary axial and tangential induced velocity, which are the $x-$ and $y-$component of $\mathrm{d}\boldsymbol{w}$ in Eq. (9), can be derived as:

$$\mathrm{d}w_x = -\frac{\Delta\Gamma}{4\pi}\frac{x_y\,\mathrm{d}s_z - x_z\,\mathrm{d}s_y}{\|\boldsymbol{x}\|^3}$$

$$= \frac{\Delta\Gamma\,\mathrm{d}s\cos\varphi}{4\pi r^2}\frac{1-(1-\frac{h}{r})\cos\left(\beta+\psi\right)}{\left[1+(1-\frac{h}{r})^2 - 2(1-\frac{h}{r})\cos\left(\beta+\psi\right) + (\beta\tan\varphi)^2\right]^{\frac{3}{2}}} \tag{10}$$

$$\mathrm{d}w_y = -\frac{\Delta\Gamma}{4\pi}\frac{x_x\,\mathrm{d}s_z - x_z\,\mathrm{d}s_x}{\|\boldsymbol{x}\|^3}$$

$$= \frac{\Delta\Gamma\,\mathrm{d}s\sin\varphi}{4\pi r^2}\frac{1-\frac{h}{r}-\cos\left(\beta+\psi\right) - \beta\sin\left(\beta+\psi\right)}{\left[1+(1-\frac{h}{r})^2 - 2(1-\frac{h}{r})\cos\left(\beta+\psi\right) + (\beta\tan\varphi)^2\right]^{\frac{3}{2}}} \tag{11}$$

In the above equations, the length of the elementary trailed vorticity arc is $\mathrm{d}s$, which is determined using Eq. (12). The variable $\beta^*$ is the generalized azimuthal angle, as proposed in the work of Pirrung et al. (2017b). The projection of $\beta^*$ into the

20 rotor plane will be the azimuthal angle $\beta$, as shown in Eq. (13).

$$\mathrm{d}s = V_{rel}^{tp}\,\mathrm{d}t = r\,\mathrm{d}\beta^* \tag{12}$$

$$\mathrm{d}\beta = \frac{v_{ip}\,\mathrm{d}t}{r} = \mathrm{d}\beta^*\cos\varphi \tag{13}$$





Recall that the near-wake part of the trailed vorticity is defined as the first quarter revolution of the wake of the own blade. Thus, the integral of the trailing functions in Eqs. (10) and (11) with the azimuthal angle $\beta$ from 0 to $\frac{\pi}{2}$ is defined as the steady-state value of the axial and the tangential near-wake induction (noted as $W_x$ and $W_y$).

$$W_x = \int\limits_{\beta=0}^{\beta=\frac{\pi}{2}} \mathrm{d}w_x \tag{14}$$

$$W_y = \int\limits_{\beta=0}^{\beta=\frac{\pi}{2}} \mathrm{d}w_y \tag{15}$$

The value of $W_x$ and $W_y$ can be calculated by directly integrating the Biot-Savart law in Eqs. (10) and (11) numerically, such as in the previous work (Li et al., 2018). However, the computationally heavy characteristic of this method is not favourable for the purpose of time-marching aeroelastic simulations. Alternatively, the steady-state induction is approximated by applying corrections to the results of some special conditions using empirical functions and pre-calculated influence coefficient tensors, which will be described in Sect. 5. In addition, the indicial function method is used for the calculation involving integration over time and the dynamic response, which will be described in Sect. 4.

## 4   Indicial function method

The numerical implementation of the lifting-line method and the coupled method requires the radial discretization of the blade. If the blade is discretized into $N$ sections, there will be $N$ calculation points and $N+1$ trailing points. The $N+1$ trailing points define $N$ line segments of bound vorticity, and the trailed vorticities emanate from these trailing points. This is a discretized approximation of the curved bound vortex line and continuous trailed vortex sheets.

For the free-wake lifting-line method that is implemented as a time-marching fashion for numerical computations, the vortex wake system is evolving and its size is growing in time. The information of the vorticities trailed and shed in the previous time steps has to be explicitly stored. For every single vortex element, there will be influence from all other vortex elements on it. For each time step, the size of the problem is in the order of $\mathcal{O}(N_{vor}^2)$, where $N_{vor}$ is the number of vortex elements. There has been intensive work to reduce the computational effort, three approaches are highlighted. Firstly, it is possible to trim the far-wake which effectively decrease the size of $N_{vor}$ (Boorsma et al., 2018). Secondly, it is possible to use computationally efficient algorithms that decrease the size of problem to $\mathcal{O}(N_{vor} \log N_{vor})$, such as particle based method: vortex-particle or particle-particle method (Rasmussen, 2011; Ramos García et al., 2018). Thirdly, it is possible to use parallel computing with graphics processing unit (GPU) to reduce the total computational time (Marten, 2020). However, the size of $N_{vor}$ is generally in the order of $10^3$ to $10^5$ larger than the number of sections $N$. This means the time-marching lifting-line method is computationally heavy even after these modifications. Therefore the method is not practical for the aeroelastic simulation of the whole design load basis (DLB) of a wind turbine, which corresponds to more than 200 hours of real-time simulation (Hansen et al., 2015; Boorsma et al., 2020).



In the near-wake model, the trailing functions in Eqs. (10) and (11) are both approximated with the sum of two exponential functions as shown in Eq. (16). The two components are decaying with the increase of the generalized angle $\beta^*$, following the exponential functions. The reason of using the generalized azimuthal angle $\beta^*$ in Eq. (16) is to account for the influence of the downwind convection velocity on the near-wake trailed vorticity length (Pirrung et al., 2017b). The two exponential terms

represent the fast and slow response of the indicial function, respectively. In Eq. (16), the parameters of $A_i$ and $b_i$ are related to the characteristics of the dynamic response. According to Beddoes (1987), the parameters of $A_1 = 1.359$, $A_2 = -0.359$, $b_1 = 1$ and $b_2 = 4$ are favourable for straight blades. Since the focus of this work is mainly on obtaining the correct steady-state induction for swept blades, the same set of parameters are used.

$$\mathrm{d}\tilde{w} = \frac{\Delta\Gamma r}{4\pi h|h|}(A_1 e^{-b_1\beta^*/\Phi} + A_2 e^{-b_2\beta^*/\Phi})\,\mathrm{d}\beta^* \tag{16}$$

Assuming $\Phi$ is constant, the approximated near-wake induction for a specific value of generalized azimuthal angle $\beta^*$ is the integral of the trailing function in Eq. (16) from $0$ to $\beta^*$.

$$\tilde{W}(\beta^*) = \frac{\Delta\Gamma r}{4\pi h|h|}\Phi\left[\frac{A_1}{b_1}(1 - e^{-b_1\beta^*/\Phi}) + \frac{A_2}{b_2}(1 - e^{-b_2\beta^*/\Phi})\right] \tag{17}$$

When the value of $\beta^*$ approaches infinity, we have the approximated steady-state near-wake induction $\tilde{W}(\beta^* = \infty)$:

$$\tilde{W}(\beta^* = \infty) = \frac{\Delta\Gamma r}{4\pi h|h|}\Phi\left(\frac{A_1}{b_1} + \frac{A_2}{b_2}\right) \tag{18}$$

In Eq. (18), since the value of $A_i$ and $b_i$ are constants and the value of $h$ and $r$ are only dependent on the geometry, the value of $\Phi$ can be interpreted as a normalized steady-state near-wake induction for unit strength of trailed vorticity. It will be used to represent the steady-state near-wake induction in the following sections.

One of the important features of the near-wake model is the use of exponential functions to approximate the trailing function that is based on the Biot-Savart law. The approximated trailing function can then be integrated using the indicial function

approach instead of using direct numerical integration. With this approach, the information of the individual trailed vortex elements emitted from the previous time steps is implicitly stored. For every time step, it is only necessary to calculate the decrement of the induction at the previous time step and the increment of the induction at the current time step.

$$\tilde{X}_w^i = \tilde{X}_w^{i-1}e^{-b_1\Delta\beta^*/\Phi} + \tilde{D}_X\Delta\Gamma(1 - e^{-b_1\Delta\beta^*/\Phi}) \tag{19}$$

$$\tilde{Y}_w^i = \tilde{Y}_w^{i-1}e^{-b_2\Delta\beta^*/\Phi} + \tilde{D}_Y\Delta\Gamma(1 - e^{-b_2\Delta\beta^*/\Phi}) \tag{20}$$

where

$$\tilde{D}_X = \frac{r}{4\pi h|h|}\frac{A_1}{b_1}\Phi \tag{21}$$

$$\tilde{D}_Y = \frac{r}{4\pi h|h|}\frac{A_2}{b_2}\Phi \tag{22}$$

The fast and slow response terms are calculated separately and then summed together to get the complete near-wake induction.

$\tilde{W}^i = \tilde{X}_w^i + \tilde{Y}_w^i \tag{23}$





The problem is now in the order of $\mathcal{O}(N^2)$ for each time step, where the number of sections $N$ is practically only 50 to 100 and is much smaller than $N_{vor}$. The computational effort is low and remains constant for every time step. The indicial function method could be interpreted in different ways, for example: first-order low-pass filter, solution of the first-order ordinary differential equation (ODE), convolution of the induction function, Duhamel's integral and exponential time differencing (ETD).

### 4.1 Distinguish the analytical and approximated induction

It could be confusing that the approximated value of the steady-state near-wake induction in Eq. (18) corresponds to $\beta = \infty$ while the analytical value of the steady-state near-wake induction in Eqs. (14) and (15) corresponds to $\beta = \frac{\pi}{2}$.

For the analytical near-wake induction $W_x$ and $W_y$ that are calculated directly from the Biot-Savart law in Eqs. (14) and (15), the integration is from $\beta = 0$ to $\beta = \frac{\pi}{2}$ because it corresponds to the first quarter revolution of the own wake. Otherwise, if integrated from $\beta = 0$ to $\beta = \infty$, the induction will correspond to the whole helical wake of the own blade until infinitely far downstream. For the integration from $\beta = \frac{\pi}{2}$ on-wards until infinity, the calculated induction belongs to the far-wake part.

For the approximated induction in Eq. (17), the integral from zero to infinity corresponds to the steady-state value of the approximated near-wake induction. Because it is only to approximate the analytical near-wake induction in Eqs. (14) and (15), and does not include the far-wake part. The relationship between the approximated and the analytical steady-state near-wake axial and tangential induction are summarized in the following equations. The negative sign in Eq. (15) is due to the definition of the positive direction of the tangential induction.

$$W_x \approx \tilde{W}_x(\beta^* = \infty) = \frac{\Delta\Gamma r}{4\pi h |h|} \Phi_x \left( \frac{A_1}{b_1} + \frac{A_2}{b_2} \right) \tag{24}$$

$$W_y \approx \tilde{W}_y(\beta^* = \infty) = -\frac{\Delta\Gamma r}{4\pi h |h|} \Phi_y \left( \frac{A_1}{b_1} + \frac{A_2}{b_2} \right) \tag{25}$$

The difference between the analytical and the approximated near-wake induction is illustrated in Fig. 3. From the left figure, it could be observed the analytical and the approximated trailing function are different and are difficult to compare. In the right figure, the integral of the trailing function representing the induction for different size of the azimuthal angle is shown. It could be observed the steady-state value of the approximated induction at $\beta = \infty$ correspond to the analytical near-wake induction at $\beta = \frac{\pi}{2}$. So, for the approximated induction function as shown in the right figure, the physical meaning of $\beta$ is not strictly the azimuthal angle. Instead, it is a measure of the time that the vortex has been emanated from the trailing point.

## 5 The steady-state value

The different methods of obtaining the normalized steady-state near-wake induction $\Phi$, in the original implementation, in the previous modifications and in the suggested modification will be described in this section. For the suggested modification, details of the modified convective correction are described. Then, the modified indicial functions are given. Finally, the algorithm of computing the induction from given geometry and vorticity strength is summarized.





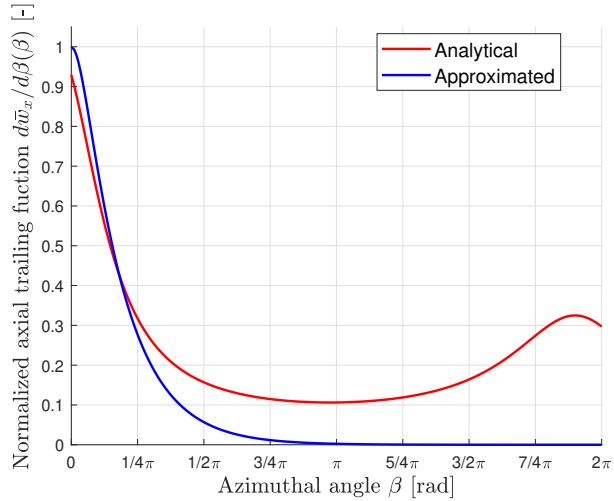
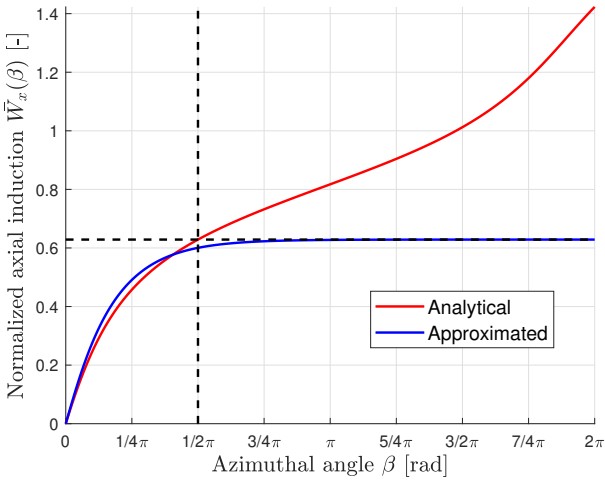

**Figure 3.** Illustration of the difference between the analytical and the approximated normalized trailing functions (left) and the corresponding induction functions (right). The parameters for the illustration are $h/r = 0.5$, $\psi = 10°$, $\varphi = 5°$. The approximated steady-state near-wake induction is when $\beta = \infty$ and corresponds to the analytical near-wake induction at $\beta = \frac{\pi}{2}$.

## 5.1 Original implementation

In the original implementation of the near-wake model by Beddoes (1987) and further extension by Wang and Coton (2001), only the axial induction is modelled. The value of $\Phi$ that represents the normalized steady-state axial induction is determined using the empirical functions:

$$5 \quad \Phi = \begin{cases} -\frac{\pi}{4}(1 + \frac{h}{2r})\ln\left(1 - \frac{h}{r}\right) & \text{if } h/r > 0 \\ \frac{\ln\left(1 - \frac{h}{r}\right)}{1.5 + \ln\left(1 - \frac{h}{2r}\right)} & \text{if } h/r < 0 \end{cases} \quad (26)$$

There are two major limitations when using the empirical functions in Eq. (26) to approximate the near-wake induction. Firstly, when the value of $h/r$ is close to 1, which corresponds to the influence of the vorticities trailed from the tip region on the root region, the approximated steady-state result from these empirical equations will deviate significantly from the analytical results. Secondly, in these empirical equations, the value of $\Phi$ is only dependent on the relative position $h/r$, but not 10 dependent on the helix angle $\varphi$. These empirical equations implicitly assume the trailed vorticity stays in the rotor plane with zero helix angle. With the increase of the helix angle, the approximated induction will gradually deviate from the analytical results and the error from these empirical equations will increase accordingly.

## 5.2 Previous modifications

There has been previous work by Pirrung et al. (2017a, b) targeted at the two issues pointed out in the previous section. Firstly, 15 the root correction is introduced to correct the value of $\Phi$ for the condition of $h/r$ close to 1. It is discovered by Pirrung et al. (2017a) that when the trailed vorticity is in-plane ($\varphi = 0$), there is a good agreement between the analytical steady-state





near-wake axial induction $W_x$ and the approximated value calculated using Eqs. (17) and (26) with $\beta = \frac{\pi}{2}$ instead of $\beta = \infty$ (here $\beta^* = \beta$ because $\varphi = 0$). Recall that the approximated steady-state near-wake induction should correspond to $\beta = \infty$ as described in Sect. 4.1, the root correction is to scale the value of $\Phi$ accordingly.

$$\Phi_C = \frac{\tilde{W}(\beta = \frac{\pi}{2})}{\tilde{W}(\beta = \infty)}\Phi = \Phi \frac{\frac{A_1}{b_1}(1 - e^{-\frac{\pi}{2}b_1/\Phi}) + \frac{A_2}{b_2}(1 - e^{-\frac{\pi}{2}b_2/\Phi})}{\frac{A_1}{b_1} + \frac{A_2}{b_2}} \tag{27}$$

Secondly, the influence of the helix angle on the near-wake induction is modelled by introducing the convective correction. The value of $\Phi$ is adjusted with the correction to approximate the steady-state induction for the general condition of an arbitrary helix angle. The corrected value of $\Phi^*$ is from a linear interpolation of the value for the special condition of in-plane trailed vorticity ($\varphi = 0$) and the special condition with straight trailed vorticity ($\varphi = \frac{\pi}{2}$). For the condition of in-plane trailed vorticity, the value of $\Phi_C$ in Eq. (27) that is with the root correction is used. For the condition of straight trailed vorticity, the value

of $\Phi_s$ is calculated from Eq. (28) (Pirrung et al., 2017b, Eq. (7)), which is an approximation of the analytical induction of a semi-infinite line vortex.

$$\Phi_s = 0.788 \left| \frac{h}{r} \right| \tag{28}$$

$$\Phi^* = k_\Phi \Phi_s + (1 - k_\Phi)\Phi_C \tag{29}$$

The weight $k_\Phi$ is calculated from the parameter of $h/r$ and $\varphi$ with empirical functions. The empirical functions rely on two

pre-calculated influence coefficient matrices that are fitted to the results from direct numerical integration. The two matrices correspond to positive and negative value of $h/r$ respectively. The empirical functions are in the form of composite functions as in Eq. (30).

$$k_\Phi = f_\varphi \left( f_{\frac{h}{r}} \left( \frac{h}{r} \right), \varphi \right) \tag{30}$$

This approach has a very low computational cost, which is crucial for the efficiency of the coupled near- and far-wake model.

The approximated steady-state axial induction of a straight blade after these corrections is having reasonably good accuracy. In addition, the near-wake part of the tangential induction is included in the modification. It is argued by Pirrung et al. (2016) that the same value of $\Phi$ can be used for the tangential induction of straight blades and will have acceptable accuracy for a small value of $|h/r|$. This is confirmed by the analytical derivations in Appendix B1.1. For the detailed description of the modified method and the pre-calculated influence coefficient matrices, the reader is referred to Pirrung et al. (2016, 2017b).

## 5.3 Suggested modification

Recall the procedures to approximate the steady-state near-wake induction in the previous modifications by Pirrung et al. (2016). Firstly, the steady-state induction of the special conditions of in-plane and straight trailed vorticity are approximated. Secondly, the approximated steady-state induction for an arbitrary helix angle $\varphi$ is obtained by applying corrections to these two special conditions. In the modification suggested in the present work, the blade sweep is considered and the definition





of the convective correction is adjusted. In addition, different equations are used to approximate the axial induction and the tangential induction.

Firstly, for the special condition of zero helix angle (in-plane trailed vorticity), modification is needed to get the correct steady-state results for the swept blades. In the original empirical equation of $\Phi$ and also the previous modification of root

correction, the blade is assumed to be straight. When the blade is swept instead of being straight, the results from the previous methods will have offsets. One possible solution is to obtain another empirical function of $\Phi$ that includes the additional variable of blade sweep angle $\psi$. Alternatively, for this special condition of in-plane trailed vorticity, the value of $W_x$ and $W_y$ in Eqs. (14) and (15) are derived analytically to be in the form of incomplete elliptic integrals, see Appendix B1. In addition, the steady-state axial and tangential induction of the special condition of straight trailed vorticity ($\varphi = \frac{\pi}{2}$) are also derived, see

Appendix B2. This means the value of $\Phi$ for both the axial and the tangential induction can be directly calculated from the analytical equations for these two special conditions. The previous empirical equations in Eq. (26) and the root correction in Eq. (27) are then not necessary.

Secondly, the idea of convective correction for the general case of an arbitrary helix angle is used but the definition is adjusted. The convective correction is now defined as the function to obtain the steady-state induction from the special condition

of in-plane trailed vorticity ($\varphi = 0$) and possibly also the special condition of straight trailed vorticity ($\varphi = \frac{\pi}{2}$). Since the steady-state induction could be represented by the value of $\Phi$ as shown in Eq. (18), the convective correction is having the form in Eq. (31). There will be separate convective correction functions for the axial and the tangential induction.

$$\Phi^* \left( \frac{h}{r}, \psi, \varphi \right) = f_{conv} \left( k_\Phi, \Phi_{ip}, \Phi_{ss} \right) \tag{31}$$

where, in turn:

$$k_\Phi = f_{k_\Phi} \left( \frac{h}{r}, \psi, \varphi \right) \tag{32}$$

### 5.4   Prerequisites of the modified convective correction

In the previous modifications by Pirrung et al. (2017b), the empirical equations for the convective correction are dependent on two variables: the relative position $h/r$ and the helix angle $\varphi$. For the current modification, there is one more design variable that is the sweep angle $\psi$. As a result, the procedure to obtain the influence coefficient tensors involves one more degree of

freedom, which is then more complicated and requires careful considerations. Three prerequisites, which are the definition of the equivalent relative position, the normalization of the sweep angle and the determination of the feasible design space, are proposed for the ease of obtaining the influence coefficient tensors.

### 5.4.1   Equivalent relative position

The relative position $h/r$ is introduced by Beddoes (1987) to represent the geometric relative position of the trailing point and

the calculation point. It is defined as the ratio of the radial distance of the trailing point and the calculation point ($r - r_{cp}$) over the radius of the trailing point $r$, which has been explained in Sect. 3. For the simplicity of the notation, the relative position





$h/r$ is noted as $\tilde{h}$ in the following of this work. When the trailing point is further outboard compared to the calculation point, $\tilde{h}$ is positive with the value between $0$ and $1$. Instead, when the trailing point is further inboard compared to the calculation point, the value of $\tilde{h}$ is negative and is not bounded. The unbounded negative value of $\tilde{h}$ can cause unnecessary difficulties when obtaining the influence coefficient tensors. In the previous work of Pirrung et al. (2017b), the data fitting for negative value of

5  $\tilde{h}$ was performed for the range of $[-4, 0)$. However, it is difficult to argue what the range of negative $\tilde{h}$ should be to cover the design space and how many grid points are needed to ensure sufficiently good results.

In order to solve this problem, the equivalent relative position $\hat{h}$ is introduced in Eq. (33) and is bounded between -1 and 1. When $\tilde{h} > 0$, its equivalent value is itself. When $\tilde{h} < 0$, the equivalent relative position is the opposite number of the value of $\tilde{h}$ when switching the location of the calculation point and the trailing point.

$$\hat{h} = \begin{cases} h/r & \text{if } h/r > 0 \\ \frac{h}{h-r} & \text{if } h/r < 0 \end{cases} \tag{33}$$

### 5.4.2 Normalization of sweep angle

Another procedure to ease the process of obtaining the influence coefficient tensors is to normalize the sweep angle $\psi$. For the induction function in Eqs. (10) and (11), the blade sweep is described by the sweep angle $\psi$, which is defined as the azimuthal difference between the calculation point and the trailing point. For a specific swept blade and when the trailing point

15  is further outboard compared to the calculation point ($\hat{h} > 0$), the range of $\psi$ will generally increase with the increase of $\hat{h}$. This is illustrated in Fig. 4 for the same calculation point but with different trailing points. When $\hat{h} < 0$, there will be a similar dependency of the range of $\psi$ on the value of $|\hat{h}|$.

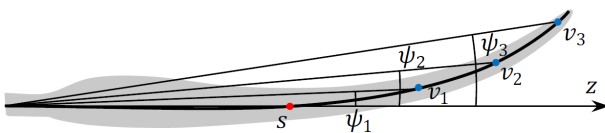

**Figure 4.** Illustration of the variation of the range of the sweep angle $\psi$ with the increase of the relative position $\hat{h}$ for a swept blade. The calculation point $s$ is further inboard compared to the trailing point and is not changed. When the trailing point is changing from $v_1$ to $v_2$ and then to $v_3$, the value of $\hat{h}$ together with the sweep angle $\psi$ are increasing.

The spread of the realistic points in the 2-D plot of $\psi$ against $\hat{h}$ will expand with the increase of $|\hat{h}|$. This will introduce difficulties when obtaining the influence coefficient tensors through data fitting. Practically, the data fitting is performed on a

20  sampling mesh grid with uniform spacing for each of the design variables and is intended to cover the whole design space. There are three design variables of $\hat{h}$, $\psi$ and $\varphi$ which corresponds to a cuboid space. Because of the dependency of the sweep angle $\psi$ on the equivalent relative position $\hat{h}$, the realistic design space inside this cuboid design space will be highly skewed. There will be many sampling grid points that correspond to unrealistic conditions. If directly using the uniformly spaced mesh grid within this design space, the data fitting will aim to minimize the error for both realistic and unrealistic conditions. This is





harmful to the quality of the fitted results, especially when the weight on the unrealistic conditions, which is measured by the number of sampling grid points that are unrealistic, is too large.

In addition, when the value of $|\hat{h}|$ is close to zero, the feasible range of $\psi$ is also small, so that the realistic conditions are clustered together into a small space inside the cuboid space for the data fitting. This means there will be insufficient number
of sampling points in this region and the fitted data can not sufficiently represent the features of the blade sweep. Then, it will be difficult to correctly approximate the steady-state induction using these fitted influence coefficients for a small value of $|\hat{h}|$. Furthermore, the data fitting for a small value of $|\hat{h}|$ is important for the calculation of the induction on the blade, because it represents the influence of the trailed vorticity on the neighbouring sections.

As a result, it is favourable to normalize the sweep angle to spread the realistic design space more evenly inside the cubic
parameter space for data fitting and also to proactively enlarge the spread of the realistic conditions for a small value of $|\hat{h}|$. The proposed method of normalizing the sweep angle $\psi$ is dividing it by $\hat{h}$. The normalized sweep angle $\tilde{\psi}$ can be considered as a measure of the blade local curvature.

$$\tilde{\psi} = \frac{\psi}{\hat{h}} \tag{34}$$

### 5.4.3 Range of feasible designs

Since the data fitting is practically performed in a cuboid parameter space, it is necessary to determine the range of each variable. For the value of $|\hat{h}|$, the range is $(0, 1)$. For the helix angle, the range is from $0$ to $\frac{\pi}{2}$. It is difficult to directly determine the range of the normalized sweep angle $\tilde{\psi}$.

To obtain the range of the normalized sweep angle, an initial numerical study is performed by calculating the value of $\hat{h}$ and $\tilde{\psi}$ for a large variety of swept blades. The planform of the swept blades used in the numerical test is obtained from a quadratic
Bézier curve which is parameterized with: sweep ratio $\bar{r}_s$, sweep magnitude $\Delta d$ and tip sweep angle $\Lambda_{tip}$, and is illustrated in Fig. 5. The quadratic Bézier curve is modified so that the exponent is able to be changed and is not necessarily being two. Then, another parameter, which is the exponent factor, is introduced so that the main-axis is possible to have more variety of local curvatures.

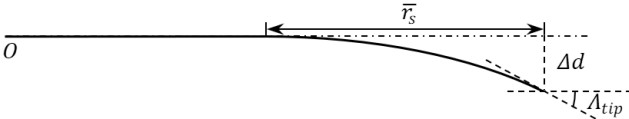

**Figure 5.** The parameterization of the swept blade with sweep ratio $\bar{r}_s$, sweep magnitude $\Delta d$ and tip sweep angle $\Lambda_{tip}$ (Li et al., 2018).

The purpose of this preliminary study is to determine the range and also the Pareto front of the design variables. So, the
range of the geometric variables for this numerical study is chosen to represent the blades with relatively large sweep. The range of the sweep ratio is from 0.25 to 0.75. The ratio of the sweep magnitude over the sweep ratio is set to vary between 0.2 and 1. So, the swept magnitude $\Delta d$ is from 20% to 100% the value of sweep ratio $\bar{r}_s$. The tip sweep angle $\Lambda_{tip}$ is varying from 25° to 57°. The exponent of the Bézier curve is varying from 1.5 to 2.5. The blade for the test is with a hub radius equal

to 2% of the rotor radius, which is relatively small when compared to typical wind turbines. The blade is discredited into 50 to 300 sections using cosine spacing. The numerical test is performed for both backward swept blades and forward swept blades. Since the scatter plot of the realistic value of $(\hat{h}, \tilde{\psi})$ is approximately symmetric with respect to the two Cartesian coordinate axes of $\hat{h} = 0$ and $\tilde{\psi} = 0$, only the first quadrant is shown in Fig. 6.

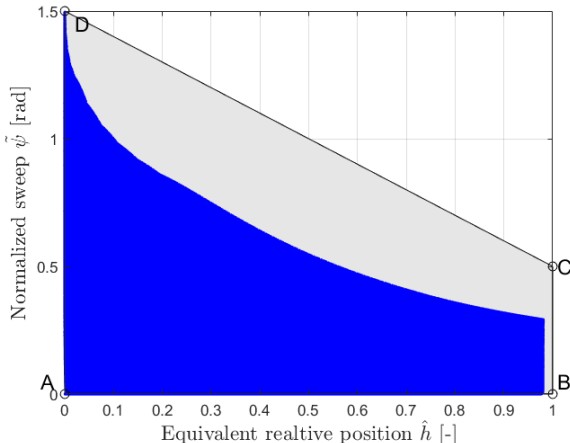

**Figure 6.** The scatter plot of the realistic conditions of the normalized sweep angle $\tilde{\psi}$ against the equivalent relative position $\hat{h}$ in the first quadrant.

The range of $\hat{h}$ is firstly investigated. From the figure, the minimum possible value of $\hat{h}$ is around $1.4 \times 10^{-5}$, and the maximum value is approximately 0.98. This gives guidelines to the range of $\hat{h}$ for the data fitting.

Secondly, according to the scatter plot in Fig. 6, it is possible to have a trapezoid region of the design variables of $\hat{h}$ and $\tilde{\psi}$ instead of a rectangular region for the first quadrant. This can reduce the ratio of the unrealistic conditions inside the design space, which is beneficial for the data fitting. The trapezoid region for the first quadrant is determined with the four corner
points of $A : (0,0)$, $B : (1,0)$, $C : (1,0.5)$ and $D : (0,1.5)$. It is possible to introduce another variable $\hat{\psi}$ to represent the blade sweep, so that it is possible to have a rectangular space of $(\hat{h}, \hat{\psi})$ that corresponds to this trapezoid design space. For the other quadrants, the trapezoid region is symmetric with the two Cartesian coordinate axes of $\hat{h} = 0$ and $\tilde{\psi} = 0$. The relationship between $\tilde{\psi}$ and $\hat{\psi}$ is given by:

$$\hat{\psi} = \frac{\tilde{\psi}}{1.5 - |\hat{h}|} = \frac{\psi}{1.5\hat{h} - \hat{h}|\hat{h}|} \tag{35}$$

**5.5   Modified convective correction**

In this section, the modified convective correction is described in detail. The idea is similar to the method of calculating the corrected value of $\Phi^*$ using empirical equations and influence coefficient matrices by Pirrung et al. (2017b).





### 5.5.1 The base trailing function and base induction

For the trailing functions of $\mathrm{d}w_x$ and $\mathrm{d}w_y$ in Eqs. (10) and (11), there are trigonometric functions of sine and cosine which are not favourable for the analytical derivation and will also impose difficulties to the practical implementation. This is because when calculating the ratio of the two values that contains sine or cosine, the issue of dividing by zero could occur. As a result, the two new trailing functions of $\mathrm{d}w_I$ and $\mathrm{d}w_{II}$ are introduced, they are noted as the base trailing functions. The trailing functions of $\mathrm{d}w_x$ and $\mathrm{d}w_y$ could be considered as the projections of the base trailing function $\mathrm{d}w_I$ and $\mathrm{d}w_{II}$ with the helix angle $\varphi$.

$$\mathrm{d}w_I = \frac{\mathrm{d}w_x}{\cos\varphi} \tag{36}$$

$$\mathrm{d}w_{II} = \frac{\mathrm{d}w_y}{\sin\varphi} \tag{37}$$

The steady-state value of the near-wake base induction corresponds to the integral of the base trailing functions in Eqs. (36) and (37) with the azimuthal angle $\beta$ from 0 to $\frac{\pi}{2}$.

$$W_I = \int_{\beta=0}^{\beta=\frac{\pi}{2}} \mathrm{d}w_I = \frac{W_x}{\cos\varphi} \tag{38}$$

$$W_{II} = \int_{\beta=0}^{\beta=\frac{\pi}{2}} \mathrm{d}w_{II} = \frac{W_y}{\sin\varphi} \tag{39}$$

The normalized base axial and tangential induction are also introduced, they are defined similar to the normalized axial and tangential induction in Eqs. (24) and (25).

$$\Phi_x = \Phi_I \cos\varphi \tag{40}$$

$$\Phi_y = \Phi_{II} \sin\varphi \tag{41}$$

For the special condition of in-plane trailed vorticity ($\varphi = 0$) and straight trailed vorticity ($\varphi = \frac{\pi}{2}$), the normalized base induction of $\Phi_{ip}$ and $\Phi_{ss}$ are derived analytically in Appendix B1 and B2.

If the shape of the blade does not change (or the change is within a threshold) between two time-steps, only the helix angle $\varphi$ will change during the convergence calculation. So that the corresponding values of $\Phi_{ip}$ and $\Phi_{ss}$ are not necessary to be re-calculated, but can be stored and reused instead.

### 5.5.2 The three-layer composite function

The convective correction is an empirical composite function of three independent variables which corresponds to three layers. These empirical functions are based on polynomial functions and rational functions. The composite functions are designed so that there is only one independent variable for each layer. Then, an optimum approach will be letting the helix angle $\varphi$ be the final layer in the composite functions.





For a given combination of the three design variables $(\hat{h}, \hat{\psi}, \varphi)$, the computation will begin from the influence coefficient tensor and the normalized sweep $\hat{\psi}$, which is the first layer. The results from the first layer will be the influence coefficients for the second layer, which is only the function of $\hat{h}$. The results from the second layer will be the coefficients for the third layer, which is only the function of $\varphi$. In this final layer, the factor of $k_\Phi$ for the convective correction is then computed. If the geometry is not changed, only the final layer associated with the helix angle $\varphi$ needs to be re-calculated during the iterations. The calculated coefficients from the first two layers of the composite function associated with the blade geometry can be saved and reused.

Following the aforementioned description, the function of the convective correction is a triple composite function that has the form as in Eq. (42).

$$k_\Phi = f_\varphi \left( f_{\hat{h}} \left( f_{\hat{\psi}} \left( \hat{\psi} \right), \hat{h} \right), \varphi \right) \tag{42}$$

The influence coefficient tensors for the axial and the tangential induction are different and will be described separately. In addition, the whole design space is divided into several sub-spaces with their own influence coefficients, which is for the ease of data fitting. The empirical functions for both the axial and tangential normalized base induction and for all the regions are the same and are as follows:

$$k_\Phi = \frac{a_{\hat{h},1}\varphi^4 + a_{\hat{h},2}\varphi^3 + a_{\hat{h},3}\varphi^2 + a_{\hat{h},4}\varphi + 1}{a_{\hat{h},5}\varphi^3 + a_{\hat{h},6}\varphi^2 + a_{\hat{h},7}\varphi + 1} \tag{43}$$

$$a_{\hat{h},i} = a_{\hat{\psi},i,1}|\hat{h}|^5 + a_{\hat{\psi},i,2}|\hat{h}|^4 + a_{\hat{\psi},i,3}|\hat{h}|^3 + a_{\hat{\psi},i,4}|\hat{h}|^2 + a_{\hat{\psi},i,5}|\hat{h}| + a_{\hat{\psi},i,6} \tag{44}$$

$$a_{\hat{\psi},i,j} = \mathbf{I}_{i,j,1}\hat{\psi}^4 + \mathbf{I}_{i,j,2}\hat{\psi}^3 + \mathbf{I}_{i,j,3}\hat{\psi}^2 + \mathbf{I}_{i,j,4}\hat{\psi} + \mathbf{I}_{i,j,5} \tag{45}$$

### 5.5.3 Influence coefficients for axial induction

For the approximation of the normalized axial induction $\Phi_I$, which is defined in Eq. (24), the whole parameter space is divided into three regions and each with its own influence coefficient tensor. The definition of the three regions for the parameter space of $(\hat{h}, \hat{\psi})$ and the corresponding influence coefficients for the axial induction are summarized in Table 1.

The first region corresponds to the first and fourth quadrant of the design space of $(\hat{h}, \hat{\psi})$. This is when the calculation point is further inboard compared to the trailing point ($\hat{h} > 0$) for both the condition of backward sweep ($\hat{\psi} > 0$) and also forward sweep ($\hat{\psi} < 0$). The influence coefficient tensor is $\mathbf{I}^{\mathbf{a1}}$. The value of $\Phi_I^*$ is calculated from the convective correction factor $k_{\Phi_I}$ and the normalized induction $\Phi_{I,ip}$.

$$\Phi_I^* = k_{\Phi_I} \Phi_{I,ip} \tag{46}$$

where $\Phi_{I,ip}$ is calculated using Eq. (B6).

The second region corresponds to the third quadrant of the design space of $(\hat{h}, \hat{\psi})$. This is when the calculation point is further outboard compared to the trailing point for the forward swept blades. The influence coefficient tensor is $\mathbf{I}^{\mathbf{a2}}$. The value of $\Phi_I^*$ is also calculated with Eq. (46).



The third region corresponds to the second quadrant of the design space of $(\hat{h}, \hat{\psi})$. This is when the calculation point is further outboard compared to the trailing point for the backward swept blades. The influence coefficient tensor is $\mathbf{I^{a3}}$. The convective correction in this region is the linear interpolation between $\Phi_{I,ip}$ and $\Phi_{I,ss}$ with the weight of $k_{\Phi_I}$.

$$\Phi_I^* = k_{\Phi_I}\Phi_{I,ip} + (1 - k_{\Phi_I})\Phi_{I,ss} \tag{47}$$

where $\Phi_{I,ip}$ is calculated using Eq. (B6) and $\Phi_{I,ss}$ is calculated using Eq. (B22).

The influence coefficient tensors of $\mathbf{I^{a1}}$, $\mathbf{I^{a2}}$ and $\mathbf{I^{a3}}$ with double-precision floating-point numbers are in the online supplement (Li et al., 2021). In addition, a version with reduced digits is in Appendix D1.

**Table 1.** The definition of the three regions for the parameter space of the equivalent relative position $\hat{h}$, the normalized sweep angle $\hat{\psi}$ and the corresponding influence coefficients for the axial induction. The equation of the convective correction and the maximum relative error of the fitted induction are also listed.

| Name | Range of $\hat{h}$ | Range of $\hat{\psi}$ | Influence coefficient | Convective correction equation | Maximum relative error |
|---|---|---|---|---|---|
| Region a1 | (0, 1) | [-1, 1] | $\mathbf{I^{a1}}$ | Eq. (46) | 0.78% |
| Region a2 | (-1, 0) | [-1, 0] | $\mathbf{I^{a2}}$ | Eq. (46) | 1.10% |
| Region a3 | (-1, 0) | [0, 1] | $\mathbf{I^{a3}}$ | Eq. (47) | 1.34% |

### 5.5.4 Influence coefficients for tangential induction

For the approximation of the normalized tangential induction $\Phi_{II}$, the whole parameter space of $(\hat{h}, \hat{\psi})$ is divided into two regions and each with its own influence coefficient tensor. The definition of the two regions for the parameter space of $(\hat{h}, \hat{\psi})$ and the corresponding influence coefficients for the tangential induction are summarized in Table 2.

The first region corresponds to the first and fourth quadrant of the design space of $(\hat{h}, \hat{\psi})$. This corresponds to when the calculation point is further inboard compared to the trailing point and for both the condition of backward sweep ($\hat{\psi} > 0$)

and also forward sweep ($\hat{\psi} < 0$). The influence coefficient tensor is $\mathbf{I^{t1}}$. The value of $\Phi_{II}^*$ is calculated from the convective correction factor $k_{\Phi_{II}}$ and the normalized base induction $\Phi_{II,ip}$.

$$\Phi_{II}^* = k_{\Phi_{II}}\Phi_{II,ip} \tag{48}$$

where $\Phi_{II,ip}$ is calculated using Eq. (B7).

The second region corresponds to the second and third quadrant of the design space of $(\hat{h}, \hat{\psi})$. This corresponds to when

the calculation point is further outboard compared to the trailing point and for both the condition of backward sweep ($\hat{\psi} > 0$) and also forward sweep ($\hat{\psi} < 0$). The influence coefficient tensor is $\mathbf{I^{t2}}$. The value of the corrected $\Phi_{II}^*$ is also calculated with Eq. (48).

As for the axial induction, the influence coefficient tensors of $\mathbf{I^{t1}}$ and $\mathbf{I^{t2}}$ for the tangential induction with double-precision floating-point numbers are in the online supplement. In addition, a version with reduced digits is in Appendix D2.



**Table 2.** The definition of the two regions for the parameter space of the equivalent relative position $\hat{h}$, the normalized sweep angle $\hat{\psi}$ and the corresponding influence coefficients for the tangential induction. The equation of the convective correction and the maximum relative error of the fitted induction are also listed.

| Name | Range of $\hat{h}$ | Range of $\hat{\psi}$ | Influence coefficient | Convective correction equation | Maximum relative error |
|---|---|---|---|---|---|
| Region t1 | (0, 1) | [-1, 1] | $\mathbf{I^{t1}}$ | Eq. (48) | 0.54% |
| Region t2 | (-1, 0) | [-1, 1] | $\mathbf{I^{t2}}$ | Eq. (48) | 0.95% |

### 5.5.5 Quality of the fitted influence coefficients

The quality of the fitted influence coefficients for the modified convective correction described in Sect. 5.5 is tested numerically in this section. The numerical test is performed on a mesh grid with very fine resolution. The results of the base induction defined in Eqs. (38) and (39) calculated from the numerical integration of the Biot-Savart law are compared with the results calculated from the convective correction. The relative error is defined in Eq. (49).

$$\epsilon = \left| \frac{W - \tilde{W}}{W} \right| \times 100\% = \left| \frac{\Phi - \Phi^*}{\Phi} \right| \times 100\% \tag{49}$$

The numerical integration is calculated using the Runge-Kutta algorithm with Dormand–Prince method implemented in the `ode45` function in MATLAB version 2020a (Shampine and Reichelt, 1997). The relative and absolute error tolerances of the numerical solver are set to $1 \times 10^{-9}$ and $1 \times 10^{-13}$, respectively. For the numerical test, the range of the helix angle is from 0 to $89.8°$ with the spacing of $0.05°$. The range of $\hat{\psi}$ is from -1 to 1 with the spacing of $1 \times 10^{-3}$. The range of $|\hat{h}|$ is from $1 \times 10^{-5}$ to 0.99. The spacing is $1 \times 10^{-5}$ for $|\hat{h}|$ between $1 \times 10^{-5}$ and $2 \times 10^{-4}$ and the spacing is $2 \times 10^{-4}$ for $|\hat{h}|$ between $2 \times 10^{-4}$ and 0.99. For each region, the maximum relative error that is defined in Eq. (49) is calculated and is summarized in Table 1 and 2. In total, for both the axial and the tangential induction, each test corresponds to $3.57 \times 10^{10}$ different conditions.

It can be seen that for both the axial and the tangential induction, the results calculated using the convective correction method with the fitted influence coefficient tensors have relatively high accuracy. In addition, for both the base axial and tangential induction, and for all regions, the relative error is always zero when $\varphi = 0$. This is because of the well-chosen empirical function in Eq. (43).

### 5.5.6 When the parameter is outside the range

The user of the coupled model should bear in mind that the model has its limitations with certain range of validity. The data fitting was performed on a relatively large range, which is intended to cover most of the swept blades. However, it is possible that the input value is outside of the range of validity. As a result, it is necessary to put a limit to the input parameters for the model to avoid catastrophic failure of the model. The range of the input variables and the corresponding physical representation are explained. Then, the limits on the input variables and their effects are described.



For the helix angle $\varphi$, the data fitting and the tests are performed on the range of $[0, 89.8°]$. When the value of $\varphi$ is less than zero, it corresponds to the trailed vorticity convects upstream. Since the trailing functions in Eqs. (36) and (37) are even functions of the helix angle $\varphi$, the absolute value of $\varphi$ should be used when $\varphi$ is less than zero. When the value of $|\varphi|$ is greater than 89.8°, it is almost equivalent to having straight trailed vorticity ($|\varphi| = 90°$). So, it is possible to put an upper boundary of

89.8° to the helix angle. For example, for the standstill condition with 90° helix angle, the value of $W_I$ and $W_{II}$ are calculated with $\varphi = 89.8°$. But when calculating $W_x$ and $W_y$ from $W_I$ and $W_{II}$, the value of $\varphi = 90°$ is used. The limiting of the helix angle will only introduce negligible error.

For the normalized relative position $\hat{h}$, the numerical test in Sect. 5.5.5 has been performed for $|\hat{h}| \in [1 \times 10^{-5}, 0.99]$. For $|\hat{h}| > 0.99$, it corresponds to the influence of the blade tip on the part of the blade that is within 1% of radius. This range can

only be reached if the user extends the blade until the rotational center, since the hub radius is mostly larger than 2% of the rotor radius. The aerodynamic load at this region is not important, so the value of $|\hat{h}|$ should be simply set to the upper limit of 0.99. For $|\hat{h}| < 1 \times 10^{-5}$, it corresponds to the influence of the trailing vorticity on the neighbouring sections when the discretization of the blade is very fine using cosine spacing with more than 300 sections. So, it is recommended to limit the number of sections to be no greater than 250.

For the normalized sweep $\hat{\psi}$, the numerical test in Sect. 5.5.5 has been performed for $\hat{\psi} \in [-1, 1]$. For the parameter study in Sect. 5.4.3, the blades with a maximum sweep angle of 57° is within this range. So, if the blade is smooth, the blade with forward or backward swept of less than 57° should be within the validity range. If the blade has a higher sweep angle, it is also possible that the condition is still within the validity range because there is some margin as shown in Fig. 6. However, if the blade has significant sweep, it is possible the normalized sweep is outside the validity range. In addition, if the blade main-axis

has *kinks* (i.e. non-continuous derivative), it is possible that there is very high value of $\hat{\psi}$ around these regions. Both conditions can cause uncertain performance of the model, so the value of $\hat{\psi}$ should be limited to the bound of $[-1, 1]$. In addition, since both conditions require attention from the user, a warning message should be printed by the computer program.

## 5.6 The modified indicial function

The indicial function described in Sect. 4 is also modified so the modified convective correction can be applied. Firstly, since

the normalized induction of $\Phi_x$ and $\Phi_y$ are having trigonometric functions of cosine and sine, their value could reach zero. If the normalized induction $\Phi$ that is used in the exponent terms in Eqs. (19) and (20) is close to zero, the indicial function will have very poor numerical performance. As a result, the base normalized induction should be used in the exponent terms instead. Secondly, the dynamic response of the axial and tangential induction are assumed to be similar. Then, the normalized axial base induction $\Phi_I$ defined in Eq. (40) is used in the exponential terms in both the axial and the tangential indicial function.

In addition, a lower limit of 0.01 is applied to the value of $\Phi_I$ to avoid the too fast response when the value of $\Phi_I$ is close to zero.





For the axial induction, the modified indicial functions are:

$$\tilde{W}_x^i = \tilde{X}_{w,x}^i + \tilde{Y}_{w,x}^i \tag{50}$$

$$\tilde{X}_{w,x}^i = \tilde{X}_{w,x}^{i-1} e^{-b_1 \Delta\beta^*/\Phi_I} + \tilde{D}_{X,x} \Delta\Gamma (1 - e^{-b_1 \Delta\beta^*/\Phi_I}) \tag{51}$$

$$\tilde{Y}_{w,x}^i = \tilde{Y}_{w,x}^{i-1} e^{-b_2 \Delta\beta^*/\Phi_I} + \tilde{D}_{Y,x} \Delta\Gamma (1 - e^{-b_2 \Delta\beta^*/\Phi_I}) \tag{52}$$

where

$$\tilde{D}_{X,x} = \frac{r}{4\pi h|h|} \frac{A_1}{b_1} \Phi_x = \frac{r}{4\pi h|h|} \frac{A_1}{b_1} \Phi_I \cos\varphi \tag{53}$$

$$\tilde{D}_{Y,x} = \frac{r}{4\pi h|h|} \frac{A_2}{b_2} \Phi_x = \frac{r}{4\pi h|h|} \frac{A_2}{b_2} \Phi_I \cos\varphi \tag{54}$$

For the tangential induction, the modified indicial functions are:

$$\tilde{W}_y^i = \tilde{X}_{w,y}^i + \tilde{Y}_{w,y}^i \tag{55}$$

$$\tilde{X}_{w,y}^i = \tilde{X}_{w,y}^{i-1} e^{-b_1 \Delta\beta^*/\Phi_I} + \tilde{D}_{X,y} \Delta\Gamma (1 - e^{-b_1 \Delta\beta^*/\Phi_I}) \tag{56}$$

$$\tilde{Y}_{w,y}^i = \tilde{Y}_{w,y}^{i-1} e^{-b_2 \Delta\beta^*/\Phi_I} + \tilde{D}_{Y,y} \Delta\Gamma (1 - e^{-b_2 \Delta\beta^*/\Phi_I}) \tag{57}$$

where

$$\tilde{D}_{X,y} = -\frac{r}{4\pi h|h|} \frac{A_1}{b_1} \Phi_y = -\frac{r}{4\pi h|h|} \frac{A_1}{b_1} \Phi_{II} \sin\varphi \tag{58}$$

$$\tilde{D}_{Y,y} = -\frac{r}{4\pi h|h|} \frac{A_2}{b_2} \Phi_y = -\frac{r}{4\pi h|h|} \frac{A_2}{b_2} \Phi_{II} \sin\varphi \tag{59}$$

## 5.7 Algorithm of computing induction using convective correction

The algorithm of computing the axial and tangential near-wake induction using the convective correction is summarized in this section. The algorithm corresponds to the calculation from the dynamic bound vorticity strength $\Gamma_{dyn}$ to the near-wake induction $W$ in the diagram by Pirrung et al. (2017a, Fig. 3).

## 6 Far-wake model and coupling method

The basis for the far-wake model is the BEM model implemented in the HAWC2 code (Madsen et al., 2020) without tip-loss correction. The effect of increased induced velocity towards the blade tip due to the trailed vorticity induction is already included in the near-wake model. Recall that the near-wake is defined as the first quarter revolution of the non-expanding helical trailed vorticity of the own blade.

The far-wake axial induction is calculated as a function of the scaled thrust coefficient (Andersen et al., 2010; Pirrung et al., 25 2016). The scaling of the thrust coefficient is based on a coupling factor that is calculated from the axial induction from the near-wake model and the reference axial induction. This reference axial induction is computed as in the regular BEM method in



---

**Algorithm 1** Overview of the convective correction method

---

**for** each blade **do**

    **for** each blade calculation point **do**

        **for** each blade trailing point **do**

            Calculate the trailing function strength $\Delta\Gamma$ at this trailing point from the bound vorticity strength of the neighbouring sections.

            Calculate the geometric variables of $h/r$ and $\psi$ from the geometry of the calculation point and the trailing point.

            Calculate the analytical value of $\Phi_{I,ip}$, $\Phi_{II,ip}$, $\Phi_{I,ss}$ and $\Phi_{II,ss}$ using Eqs. (B12), (B13), (B22) and (B23).

            Calculate the normalized relative position $\hat{h}$ and normalized sweep angle $\hat{\psi}$ using Eqs. (33) and (35).

            Determine if the design variable $\hat{h}$, $\hat{\psi}$ and helix angle $\varphi$ are in the feasible region, following the description in Sect. 5.5.6.

            Determine the region for the axial and tangential induction based on the value of $(\hat{h},\hat{\psi})$ and choose the corresponding influence coefficient tensor following the description in Sect. 5.5.3 and 5.5.4

            Calculate the convective correction factor $k_\Phi$ using Eq. (43).

            Calculate $\Phi_I^*$ and $\Phi_{II}^*$ that is after the convective correction, Eqs. (46) to (48), following the description in Table 1 and 2.

            Calculate $X_{w,x}$, $Y_{w,x}$, $X_{w,y}$ and $Y_{w,y}$ using Eqs. (51), (52), (56) and (57).

            Calculate the contribution of the near-wake helical trailed vorticity to the axial and tangential induction of this section at the new time-step using Eq. (50) and (55).

        **end for**

        Sum the contribution of the trailed vorticities from all trailing points to the induction at this calculation point.

        Include the curved bound vortex influence, see Li et al. (2020).

    **end for**

**end for**

---

the HAWC2 code, which includes the tip-loss correction (Andersen et al., 2010; Pirrung et al., 2016). The aim of the coupling factor is that the thrust of the rotor calculated from the coupled near- and far-wake model is at a similar level as that from the reference BEM model. The scaling factor is calculated from the rotor-averaged axial induction with the weight of the annulus area, and it is applied to the far-wake axial and tangential induction. The scaling factor is set to be less than one to avoid

5    exaggerated axial induction.

    For the case of straight blades, previous studies have illustrated that the coupling factor is able to be automatically adjusted during the computation. Indeed, the dynamic response of the coupled model shows improved agreement with higher-fidelity models and experiments, when compared to the BEM method (Pirrung et al., 2017a; Schepers et al., 2021). However, the current method of coupling the near- and far-wake model is implicitly based on the assumption that the blade is straight and

10    the rotor is planar. This is because in the reference BEM, a relationship between the axial induction and the thrust coefficient that is fitted to actuator disc simulations is used (Madsen et al., 2020). When the blade is swept, the relationship between the axial induction and the thrust coefficient should differ from the case of the straight blade, especially near the blade tip. If using the same coupling method, the total thrust coefficient could have large deviations comparing to the straight blade. This means that the current coupling method is not strictly suitable for the rotors with swept blades.





For the application of the steady-state aerodynamic load calculation of swept blades under uniform inflow that is perpendicular to the rotor plane, it is also possible to fix the coupling factor equal to that of the baseline straight blade. As will be described in Sect. 8.1, the influence of blade sweep on the far-wake should be small. As a result, it is reasonable to assume the far-wake of the swept blade begins from the same position as that of the straight blades, which means using the same coupling

factor. However, the method of fixing the coupling factor is not applicable to the dynamic response calculation. The results of the coupled method with both automatically adjusted coupling factor and the fixed coupling factor will be shown in Sect. 8.

## 7   Models used for comparison

In order to assess the performance of the proposed coupled near- and far-wake model, the results from two higher-fidelity aerodynamic models are used for the comparison. In particular, a version of the lifting-line method implemented in the MIRAS

code (Ramos-García et al., 2016; Li et al., 2020) and the in-house Navier-Stokes solver EllipSys3D (Michelsen, 1992, 1994; Sørensen, 1995) are used.

In the lifting-line method used for comparison, the bound vorticity is represented by the concentrated lifting-line that is located at the quarter-chord line of the blade. This is where the trailed vortices emanate from and will form the helical vortex wake system. The induced velocity due to the trailed vorticities is evaluated at the quarter-chord line, with a possible contribu-

tion from the shed vorticity in the unsteady case. The influence of the curved bound vortex is modelled by adding the difference of the induced velocity due to the 3-D bound vorticity and an imaginary 2-D bound vorticity (infinitely long line vortex) evaluated at the three-quarter-chord point to the induction of the blade section. This implementation of the lifting-line method is labelled as LL-test in the previous work of Li et al. (2020). The coupled near- and far-wake model proposed in the present work is considered as an approximation of this implementation of the lifting-line method. So, the result from this lifting-line method

is a benchmark of how the proposed coupled method performs. In addition, the coupled method is not expected to perform better than the lifting-line method.

Apart from the lifting-line method, the results from a rotor-resolved Navier-Stokes solver were also used for comparison. The in-house finite volume code EllipSys3D solves the incompressible Navier-Stokes equation on a structured grid. Several approaches are available in EllipSys3D for dealing with turbulence. In the present study, the RANS formulation in combination

with the k-$\omega$ SST turbulence model was used (Menter, 1994).

The modified coupled near- and far-wake model is implemented in a test version of the in-house aero-servo-elastic simulation tool HAWC2 based on the release version 12.8 (Larsen and Hansen, 2007). The modifications of the near-wake model proposed in this work as well as the influence of curved bound vortex proposed by Li et al. (2020) are implemented. The implementations of the far-wake model, the coupling method as well as the iteration relaxation method are identical with the previous work

by Pirrung et al. (2017a).

The BEM method implemented in the HAWC2 code version 12.8 is also used for the comparison (Madsen et al., 2020). The BEM method is the most commonly used low-fidelity aerodynamic model. The result from the BEM method is considered as a baseline and is to illustrate the improvements of the proposed coupled method comparing to it.



# 8   Results

In this section, the aerodynamic loads calculated from different models are compared. The blades are assumed to be stiff, which means the effect of elastic deformation is not included.

## 8.1   The consistent definition of the loads

In the previous work of Li et al. (2018), the aerodynamic loads calculated from the BEM method and an early version of the coupled model are compared with the results from CFD. In that previous work, the out-of-plane loads from the coupled model and the BEM method are having similar trends but are very different from the prediction from CFD. In that previous work, it was argued that the wrong pattern of the out-of-plane load offset is due to the insufficient far-wake BEM model in the coupled model. Since the BEM method predicts the wrong pattern, the error is inherited to the coupled method because a far-wake

BEM model is used.

The previous argument is erroneous and will be illustrated using the vortex theory. It has been described in Sect. 2 that the BEM method without tip correction is equivalent to modelling the wake with concentric vortex cylinders that begin at the rotor plane. So, the far-wake BEM method with scaled inductions can be considered as having the vortex cylinders begin further downstream compared to the rotor plane. The influence of the blade sweep on the vortex wake is the in-plane shifted position of where the trailed vorticity begins. This means the influence of the blade sweep on the wake is mainly on the part of the wake

that is close to the rotor plane, so the influence on the stream-wise location where the far-wake vortex cylinders begin is very small. As a result, the corresponding influence of the far-wake reflected on the loads should not be that pronounced to have such big offsets as shown in the previous work of Li et al. (2018).

Instead, the reason is discovered to be the inconsistent definition of the loads. Recall the procedures to obtain the aerodynamic

loads in the lifting-line-like methods that rely on 2-D airfoil data, such as the BEM method, the lifting-line method and the coupled near- and far-wake model. For each blade section, the 3-D velocity at the calculation point consists of the induced velocity, the blade motion and the onset flow, and is projected into the 2-D airfoil section. After subtracting the 2-D bound vorticity induction at this section, the angle of attack and the relative velocity are calculated from the velocity triangle. Then, the 2-D lift and drag force can be calculated and are projected with respect to the rotor plane to obtain the in-plane and out-

of-plane loads. The resulting aerodynamic loads should correspond to force per unit length of curved blade length, since they are from the 2-D aerodynamic loads. If we want to have other definitions of the load, we have to multiply the load with the corresponding scaling factor. For example, to get the loads with the definition of force per unit radius, the factor $\frac{\mathrm{d}s}{\mathrm{d}r}$, which is the ratio of the local elementary increase of curved blade length over the elementary increase of radius, should be applied (Madsen et al., 2020).

In this work, the in-plane and out-of-plane loads are defined as force per unit length of $z$-coordinate, which corresponds to the radius of the straight blade. So, the factor $\frac{\mathrm{d}s}{\mathrm{d}z}$, which is the ratio of the local elementary increase of curved blade length over the elementary increase of $z-$coordinate, should be applied. In this work, the aerodynamic loads calculated from CFD is also with the same definition. The post-processing of the CFD results is done by performing planar cuts that are perpendicular



to the $z$-axis, and then integrating the pressure and viscous force along the cut contour. The results were averaged over the last 350 iterations, in order to provide mean values for the loads of the inboard part of the blade (where shedding is expected).

## 8.2 The blades for comparison

The wind turbine blades that are used for the comparison are modified based on the IEA-10.0-198 10 MW reference wind

turbine (RWT) (Bortolotti et al., 2019). The baseline straight blade is modified by aligning the half-chord line to a straight main-axis. The rotor diameter is 198 m, of which the hub radius is 2.8 m and the blade length is 96.2 m. For the swept blades, the planform is obtained from a modified Bézier curve which is parameterized with: sweep ratio $\bar{r}_s$, sweep magnitude $\Delta d$ and tip sweep angle $\Lambda_{tip}$, which has been illustrated in Fig. 5. For a clean comparison, the pre-bend as well as the blade cone are removed for all blades. The airfoils are aligned perpendicular to the curved main-axis of the half-chord line. The chord and

twist distribution of the swept blade remains the same as the baseline blade, for the sections with the same $z$-coordinate. For the baseline straight blade, the $z$-coordinate is equivalent to the radial position. For the swept blade, the length in the $z$-coordinate remains the same as the baseline straight blade. The actual radius of the swept blade is increased compared to the baseline straight blade. The backward swept blades used in this study are having the same parameters as Blade-1 to Blade-4 in the previous work of Li et al. (2018). The parameters of the four backward swept blades used for the comparison in this work are

summarized in Table 3. The sketch of the geometry of the backward swept blades and the baseline straight blade are shown in Fig. 7. In addition, four forward swept blades with the name of Blade-5 to Blade-8 that have the same parameters as the backward swept blades Blade-1 to Blade-4 but with different direction of sweep are introduced.

The operational condition is the same as the in previous work by Li et al. (2020), with uniform inflow of 8 m s$^{-1}$ perpendicular to the rotor. The rotor is operating at rotational speed of 0.855 rad s$^{-1}$, which corresponds to a tip-speed-ratio of 10.58 for

the rotor with baseline straight blades. The blades are not pitched, so the main-axis of the swept blades and the straight blade will always stay in the rotor plane.

**Table 3.** The parameters of the planforms of four backward swept blades (Li et al., 2018) .

| Name | Sweep ratio $\bar{r}_s$ | Sweep magnitude $\Delta d$ | Tip sweep angle $\Lambda_{tip}$ |
|---|---|---|---|
| Blade-1 | 50% | 10% | 20° |
| Blade-2 | 50% | 10% | 40° |
| Blade-3 | 25% | 5% | 20° |
| Blade-4 | 25% | 5% | 40° |





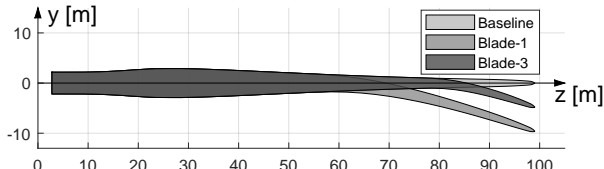
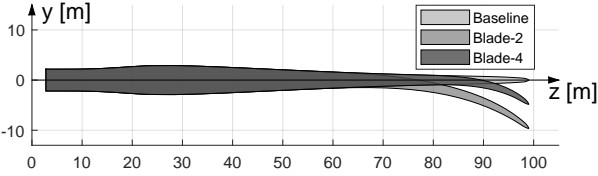

**Figure 7.** The top view of the backward swept blades Blade-1 to Blade-4 together with the baseline straight blade.

### 8.3 Description of the simulation setup

A set of rotor-resolved meshes were used for the CFD simulations, each of them corresponding to a different blade geometry. They were generated in two consecutive steps, that were fully scripted in order to ensure a similar resulting grid quality. Firstly, a structured mesh of the blade surface was generated with the openly available Parametric Geometry Library (PGL) tool (Zahle,

2019). A total of 128 cells were used in the spanwise direction, and the chordwise direction was discretized with 256 cells (with 8 of them lying on the trailing edge). Secondly, the surface mesh was radially extruded with the hyperbolic mesh generator Hypgrid (Sørensen, 1998) to create a volume grid. A total of 256 cells were used in this process, and the resulting outer domain was located at approximately 11 rotor diameters. A boundary layer clustering was taken into account, with an imposed first cell height of $1 \times 10^{-6}$ m. The resulting volume mesh accounted for a total of 14.2 million cells. An inlet/outlet strategy was

followed for the boundary conditions of the outer limit of the CFD domain, and the flow was assumed to be fully turbulent.

For the lifting-line method, each time step corresponds to $1.5°$ of azimuthal angle and each simulation is calculated for 20 thousand time steps, which correspond to 83.3 revolutions. The vortex core size is 0.1% of the local chord length. Each blade is discretized radially into 50 sections with cosine spacing. The airfoil data is from 2-D fully turbulent CFD results (Bortolotti et al., 2019). The first row of trailed vorticities begins from the lifting-line that is located at the quarter-chord line.

For the modified coupled near- and far-wake model and the BEM method implemented in the HAWC2 code, each time step corresponds to 0.01 s and each simulation is calculated for 600 s. Each blade is discretized radially into 80 sections. The same set of airfoil data that is from 2-D fully turbulent CFD result is used. For the computation of the swept blades, the coupling factor is either automatically adjusted or fixed to the value of the baseline straight blade, as described in Sect. 6. Both results for the swept blades will be shown.

### 8.4 Results for baseline geometry

Firstly, the loads of the baseline straight blade calculated from the BEM method, the modified coupled model (NW), the lifting-line method (LL) and the Navier-Stokes solver (CFD) are compared in Fig. 8. To be noted, the loads plotted from all four models are corresponding to aerodynamic force per unit length of the $z-$coordinate (equals to radius for the straight blade).

For the out-of-plane loads, the results from all the models are having good agreement. At $z-$coordinate of 80 m that corresponds to approximately 80% span, the relative difference of the out-of-plane load from the BEM method is 1.6% and 0.2% compared to CFD and LL. At the same spanwise location, the relative difference of the out-of-plane load from the coupled





method is 1.1% and 0.4% compared to CFD and LL. For the in-plane loads, the results have some small differences but are still similar. At the $z-$coordinate of 80 m, the relative difference of the in-plane load from the BEM method is 6.8% and 0.8% compared to CFD and LL. And the relative difference of the in-plane load from the coupled method is 4.3% and 1.6% compared to CFD and LL at the same spanwise location.

The differences between the CFD and LL are assumed to be related to the 2-D airfoil aerodynamic coefficients retrieved from the look-up table involved in the lifting-line approach. This source of disagreement is also to be considered for BEM and for the coupled method. The relative difference of the loads calculated from BEM and the coupled method compared to the loads from LL is relatively small. This means both the BEM and the coupled method can be used in the design optimization of a straight blade with acceptable accuracy.

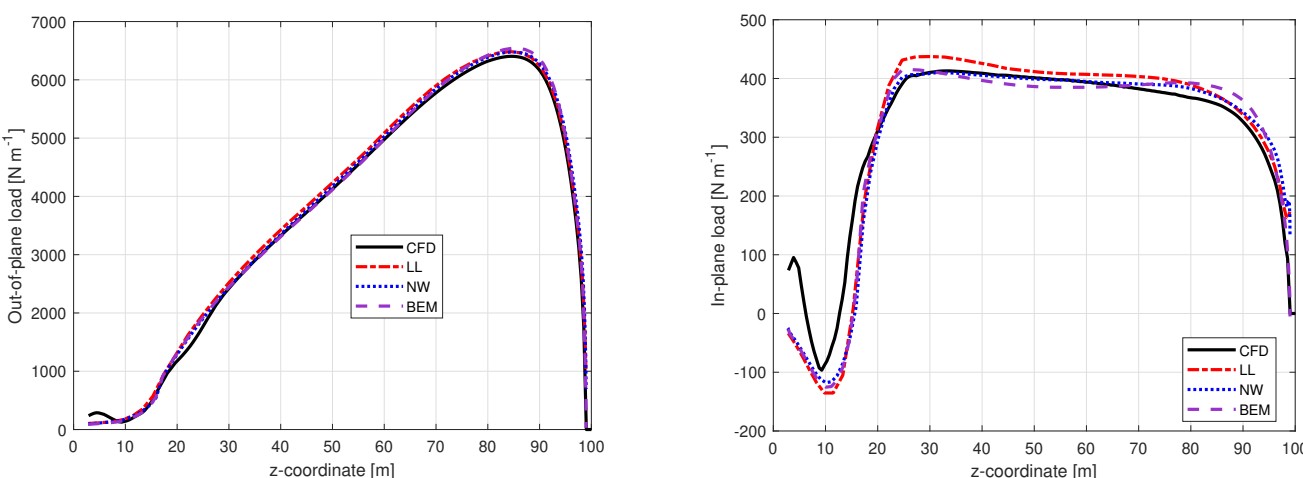

**Figure 8.** Comparison of out-of-plane load (left) and in-plane load (right) of the baseline straight blade calculated from the Navier-Stokes solver (CFD), the lifting-line method (LL), the proposed coupled method (NW) and the blade element momentum method (BEM).

**8.5    Results for backward swept blades**

The steady-state results of the swept blades are also calculated from the BEM method, the modified coupled model, the lifting-line models and the CFD. In order to clearly show the influence of the backward sweep on the loads, the difference between the loads of the backward swept blade Blade-1 with respect to the baseline straight blade is shown in Fig. 9. It is calculated by subtracting corresponding sectional loads at the same $z-$coordinate. The loads are with the definition of force per unit length

of $z-$coordinate. In this study, the focus is on the influence of blade sweep on the loads. The root region that has $z-$coordinate less than 20 m is experiencing separation and is not the focus of this study.

For both out-of-plane and in-plane load of the backward swept blade, the results from the coupled method of either automatically adjusted or fixed coupling factor are very similar. For the offset of the out-of-plane load, the result from the coupled method is in good agreement with the lifting-line method. The results are also in harmony with the result from CFD. For the

inboard part of the swept blade that the main-axis is still straight, the out-of-plane load of the swept blade is almost identical





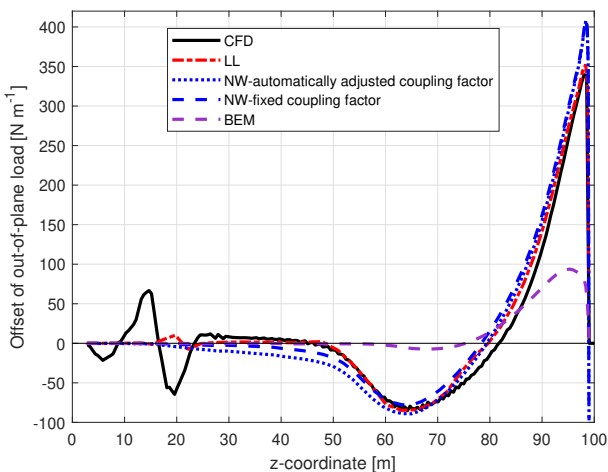
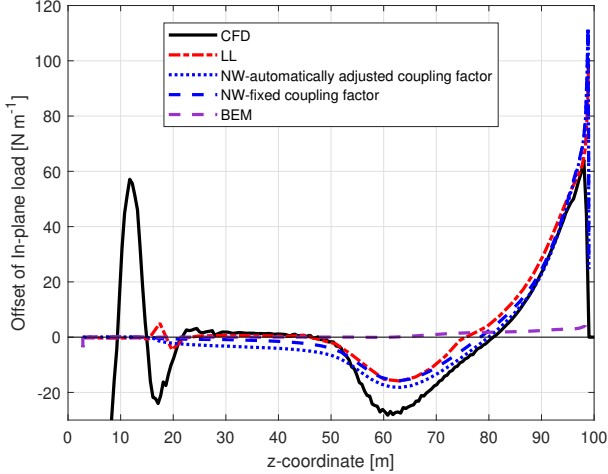

**Figure 9.** Comparison of the difference between the out-of-plane load (left) and the in-plane load (right) of the backward swept Blade-1 with respect to the baseline blade calculated from the Navier-Stokes solver (CFD), the lifting-line method (LL), the proposed coupled method (NW) and the blade element momentum method (BEM).

to that of the baseline straight blade. When moving towards the blade tip, the out-of-plane load of the swept blade is lower compared to the baseline straight blade until approximately halfway until the blade tip. Then, when moving further towards the tip, the load of the swept blade is higher compared to the baseline straight blade until almost all the way until the blade tip. This pattern was also observed in the previous work (Li et al., 2020). For the offset of the in-plane load, the result from

the coupled method is also in good agreement with the lifting-line method. Both methods can correctly predict the spanwise pattern of in-plane load redistribution of the swept blade, which is similar to the pattern seen for the out-of-plane load. Both methods underestimate the decrease of the load of the swept blade compared to CFD near $z-$coordinate of 60 m. In general, the results from the lifting-line method and the coupled method are in good agreement with CFD.

The BEM method is not able to correctly predict this pattern of the radial redistribution of the loads. For the out-of-plane

load, it predicts an maximum increase of the load near the blade tip of approximately $100\,\mathrm{N\,m^{-1}}$, while LL and CFD predicts more than $340\,\mathrm{N\,m^{-1}}$ of load increase. In addition, the BEM method is not able to predict the approximately $80\,\mathrm{N\,m^{-1}}$ decrease of the out-of-plane load at near $z-$coordinate of 65 m, as seen in the prediction by LL and CFD. For the in-plane load, the BEM method predicts that the load of the swept blade and the straight blade are almost identical along the span.

The results of the other backward swept blades are shown in Appendix C1. For all four backward swept blades, the per-

formance of the modified coupled model with either fixed or automatically adjusted coupling factor is almost as good as the lifting-line method, and both are in good agreement with CFD. In addition, an early version of the modified coupled method that has slightly lower accuracy and a smaller range of validity has been intensively used for the aeroelastic design optimization and load calculation of backward swept blade tips by Barlas et al. (2021). In that work, the proposed method with automatically adjusted coupling factor performed well for the optimization and had generally good agreement with higher-fidelity models.

This means the suggested coupled model with the current far-wake model and the automatically adjusted coupling factor is





applicable to backward swept blades if special care is taken by the user. The coupled method is having similar performance as the lifting-line method, which means it is favourable for the load calculation and design optimization of swept blades. Instead, the BEM method is not able to correctly predict the influence of the blade sweep on the loads. The poor performance of the BEM method is as expected because the influence of the curved bound vortex and the shifted starting position of the trailed

vorticity are not modelled. The results also indicate that the BEM method is not suitable for the design optimization of blades with noticeable backward sweep.

### 8.6   Results for forward swept blades

The difference between the loads of the forward swept blade Blade-5 with respect to the baseline straight blade is shown in Fig. 10. As for the backward swept blades, the loads are with the definition of force per unit length of $z-$coordinate.

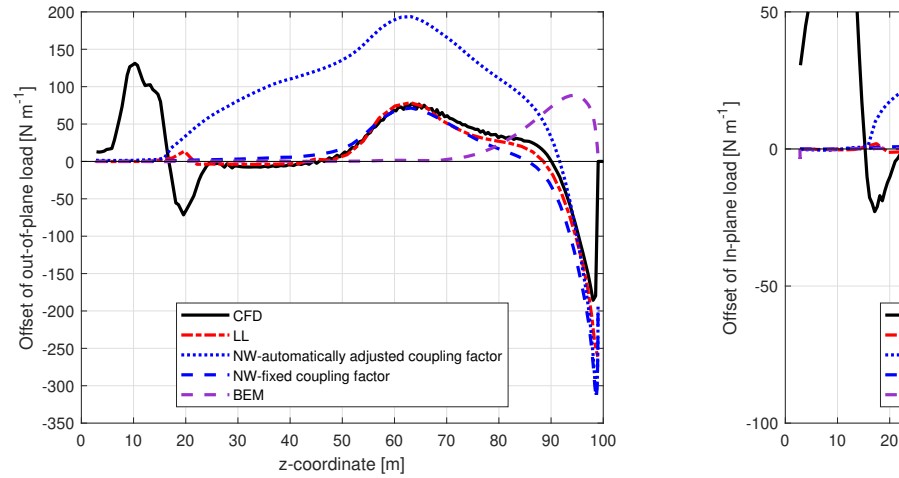
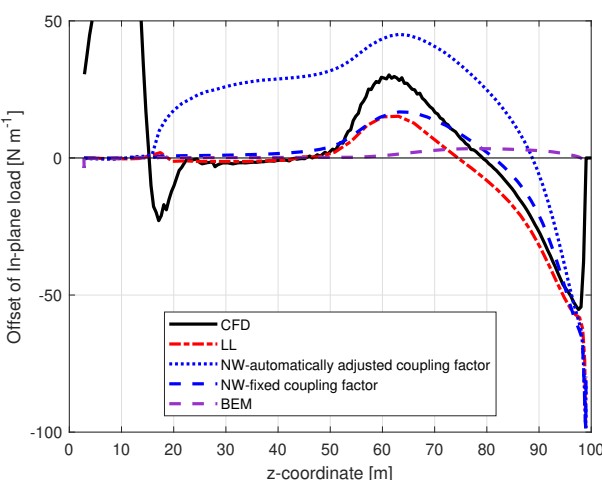

**Figure 10.** Comparison of the difference between the out-of-plane load (left) and the in-plane load (right) of the forward swept Blade-5 with respect to the baseline blade calculated from the Navier-Stokes solver (CFD), the lifting-line method (LL), the proposed coupled method (NW) and the blade element momentum method (BEM).

For the coupled method with fixed coupling factor, the results of both out-of-plane load and in-plane load are in good agreement with the higher-fidelity lifting-line method and CFD. However, for the coupled method with automatically adjusted coupling factor, the loads have significant offsets compared to the higher-fidelity models. This means the current coupling method is not capable of correctly adjusting the coupling factor automatically.

Similar to the backward swept blade cases, the BEM method is not able to predict the radial redistribution of the loads, but

predicts an increase of the load compared to the baseline straight blade near the blade tip. For the in-plane load, the BEM method predicts that the in-plane load of the swept blade and the straight blade are almost identical along the span.

The results of other forward swept blades are shown in Appendix C2. As seen for the backward swept blades, for all four forward swept blades, the performance of the modified coupled model with fixed coupling factor is almost as good as the lifting-line method, and both are in good agreement with CFD. The BEM method, on the other hand, is not able to correctly





predict the influence of the blade forward sweep on the loads. The loads predicted by the coupled method with automatically adjusted coupling factor show significant offsets compared to higher-fidelity models and should thus not be used. This means for forward swept blades, the current coupled method is only applicable to steady-state load calculation with fixed coupling factor. As a result, the current coupled method is not applicable to the aeroealstic calculation of forward swept blades.

**9   Conclusions and future work**

A computationally efficient modified coupled near- and far-wake engineering aerodynamic model for the swept wind turbine blades is proposed. The core of the modifications in this work is to obtain the steady-state induction of the near-wake, which is defined as the first quarter revolution of the helical trailed vorticity of the own blade. To achieve this, an engineering approach that combines analytical solutions and approximations based on pre-calculated influence coefficient tensors is proposed. The

far-wake model is currently based on a far-wake BEM method. The near- and far-wake model are coupled with a coupling factor that is to scale the far-wake induction, so that the thrust of the whole rotor is similar to that calculated from the BEM method. For the calculation of the steady-state condition with the uniform inflow applied perpendicular to the rotor plane, a fixed coupling factor that is determined according to the baseline straight blade can be applied.

The modified model is used to calculate the steady-state loads of the baseline straight blade, four backward swept blades

and four forward swept blades that are modified based on the IEA-10.0-198 10 MW reference wind turbine. The influence of the blade sweep on the loads predicted by the proposed method is shown to have good agreement with the prediction from higher-fidelity models, which are a version of the lifting-line solver and a Navier-Stokes solver. The numerical comparison shows that the BEM method is not able to correctly model the influence of blade sweep and is having large discrepancies with the results from the two higher-fidelity models. The improvement of the proposed coupled method over the BEM method is

significant and the results from the proposed method are having similar performance as the lifting-line method. The proposed method is computationally efficient and favourable for the application of wind turbine aero-servo-elastic simulations and design optimization. The method shows improved agreement with higher-fidelity models compared to the conventional BEM method when the model is carefully used. However, the current coupling method is not suitable for aeroelastic calculation of forward swept blades. Further work on the far-wake model and the coupling method is needed for the method to be confidently used in

the aeroelastic simulations for general swept blades.

There are several future works needed to further improve the model. Firstly, it is favourable to also have the parameters representing the dynamics of the indicial functions fitted to numerical results. This can improve the dynamic response of the coupled model. The dynamic response of swept blades from the coupled model should also be compared with results from higher-fidelity models. Secondly, using the method of fixing the coupling factor for forward swept blades reflects the limitation

of the current far-wake BEM model. It may be favourable to use the vortex cylinder model as the far-wake model instead. If so, a new method to couple the near-wake model and the far-wake model with a new definition of the coupling factor is needed. Thirdly, it could be useful to have the model further modified for the application of blades with both in-plane and out-of-plane shapes. This will also require the use of the vortex cylinder model as the far-wake model, which has the potential to model the



aerodynamic effects of the blade out-of-plane shapes. Finally, it is beneficial to investigate further possible improvements to the lifting-line method for the application of curved wind turbine blades. Then, the coupled near- and far-wake model can be improved according to it. One example is the modelling of the radial viscous drag force, especially for the swept blades.

*Data availability.* A repository (Li et al., 2021) contains the influence coefficients with double-precision floating point accuracy for the
calculation of the convective correction. In addition, a version with reduced digits is in Appendix D1.

## Appendix A: Nomenclature

$a_{\hat{h}}, a_{\hat{\psi}}$  intermediate coefficients for the convective correction

$A_1, A_2, b_1, b_2$  coefficients for the indicial functions

$\Delta d$  sweep magnitude

$\tilde{D}_X, \tilde{D}_Y$  factors for the fast and slow response in the indicial function

$\overline{G}$  indefinite integral of the normalized induction function

$h$  distance between calculation point and trailing point

$\tilde{h}$  relative position

$\hat{h}$  equivalent relative position

$\mathbf{I}$  influence coefficient tensor

$k_\Phi$  convective correction factor

$r$  radius of the trailing point

$r_{cp}$  radius of the calculation point

$\bar{r}_s$  sweep ratio

$\mathrm{d}\boldsymbol{s}$  elementary trailed vortex filament

$\Delta t$  elapsed time

$U_\infty$  wind speed

$V_{rel}^{tp}$  relative velocity of the trailing point

$v_{oop}$  out-of-plane velocity

$v_{ip}$  in-plane velocity

$\mathrm{d}w_x, \mathrm{d}w_y$  elementary axial and tangential induced velocity

$W_x, W_y$  axial and tangential near-wake induced velocity

$\tilde{W}_x, \tilde{W}_y$  approximated axial and tangential near-wake induced velocity

$\overline{W}$  normalized near-wake induced velocity

$\boldsymbol{x}$  relative position vector, pointing from the elementary trailed vorticity to the calculation point

$\tilde{X}_w, \tilde{Y}_w$  fast and slow response term of the normalized induced velocity





**Greek letters**

$\beta$     azimuthal angle of trailed vorticity

$\beta^*$     generalized azimuthal angle

$\Delta\beta^*$     change of generalized azimuthal angle in a time step

$\Delta\Gamma$     trailed vorticity strength

$\epsilon$     relative error

$\Lambda_{tip}$     tip sweep angle

$\varphi$     helix angle

$\Phi$     normalized steady-state near-wake induction

$\psi$     sweep angle

$\tilde{\psi}$     normalized sweep angle

$\hat{\psi}$     modified normalized sweep angle

$\Omega$     rotor speed

**Subscripts**

$I$     the base value of the axial induction

$II$     the base value of the tangential induction

$x$     in the axial direction

$y$     in the tangential direction

$X$     the fast response term

$Y$     the slow response term

$ip$     in-plane

$oop$     out-of-plane

$s$     straight vortex

$ss$     stand-still condition

$C$     with the root correction

$i, j$     indices of the coefficients

**Superscripts**

$*$     the value after convective correction

$i$     at time step $i$

$tp$     trailing point

$a$     axial direction

$t$     tangential direction



## Appendix B: The analytical solution of trailed functions

The analytical solutions for the two special conditions of in-plane trailed vorticity and straight trailed vorticity are derived. They correspond to the lower and upper limit of the helix pitch angle $\varphi$, which are 0 and $\frac{\pi}{2}$.

### B1 In-plane trailed vorticity

For the special condition of in-plane trailed vorticity ($\varphi = 0$), the elementary trailed vortex length $\mathrm{d}s$ is then:

$$\mathrm{d}s = r\,\mathrm{d}\beta^* = r\,\mathrm{d}\beta \tag{B1}$$

Inserting Eq. (B1) together with the condition of $\varphi = 0$ into the base trailing function in Eqs. (36) and (37), we have the base trailing function for the condition of in-plane trailed vorticity. Here the subscript of $ip$ represent in-plane trailed vorticity.

$$\mathrm{d}w_{I,ip} = \frac{\Delta\Gamma}{4\pi r} \frac{1 - (1 - \frac{h}{r})\cos(\beta + \psi)}{\left[1 + (1 - \frac{h}{r})^2 - 2(1 - \frac{h}{r})\cos(\beta + \psi)\right]^{\frac{3}{2}}}\,\mathrm{d}\beta \tag{B2}$$

$$\mathrm{d}w_{II,ip} = \frac{\Delta\Gamma}{4\pi r} \frac{1 - \frac{h}{r} - \cos(\beta + \psi) - \beta\sin(\beta + \psi)}{\left[1 + (1 - \frac{h}{r})^2 - 2(1 - \frac{h}{r})\cos(\beta + \psi)\right]^{\frac{3}{2}}}\,\mathrm{d}\beta \tag{B3}$$

The integrals of the base induction functions in Eqs. (38) and (39) with $\beta$ from 0 to $\frac{\pi}{2}$, which corresponds to the near-wake steady-state induction, are as follows.

$$W_{I,ip} = \frac{\Delta\Gamma r}{4\pi h|h|} \int_0^{\frac{\pi}{2}} \frac{h|h|}{r^2} \frac{1 - (1 - \frac{h}{r})\cos(\beta + \psi)}{\left[1 + (1 - \frac{h}{r})^2 - 2(1 - \frac{h}{r})\cos(\beta + \psi)\right]^{\frac{3}{2}}}\,\mathrm{d}\beta \tag{B4}$$

$$W_{II,ip} = \frac{\Delta\Gamma r}{4\pi h|h|} \int_0^{\frac{\pi}{2}} \frac{h|h|}{r^2} \frac{1 - \frac{h}{r} - \cos(\beta + \psi) - \beta\sin(\beta + \psi)}{\left[1 + (1 - \frac{h}{r})^2 - 2(1 - \frac{h}{r})\cos(\beta + \psi)\right]^{\frac{3}{2}}}\,\mathrm{d}\beta \tag{B5}$$

For the simplicity of the notation, the steady-state base inductions are normalized and are as follows.

$$\overline{W}_{I,ip} = \frac{W_{I,ip}}{\frac{\Delta\Gamma r}{4\pi h|h|}} = \Phi_{I,ip}\left(\frac{A_1}{b_1} + \frac{A_2}{b_2}\right) \tag{B6}$$

$$\overline{W}_{II,ip} = \frac{W_{II,ip}}{\frac{\Delta\Gamma r}{4\pi h|h|}} = -\Phi_{II,ip}\left(\frac{A_1}{b_1} + \frac{A_2}{b_2}\right) \tag{B7}$$

The indefinite integral corresponding to the definite integral of $\overline{W}_{I,ip}$ and $\overline{W}_{II,ip}$ are noted as $\overline{G}_{I,ip}$ and $\overline{G}_{II,ip}$. The two indefinite integrals are derived to be in the form of incomplete elliptic integrals.

$$\overline{G}_{I,ip}(\frac{h}{r}, \psi, \beta) = \frac{(\frac{h}{r})^2}{2 - \frac{h}{r}} E\left(\frac{\beta + \psi}{2} \,\middle|\, \frac{-4(1 - \frac{h}{r})}{(\frac{h}{r})^2}\right) + \frac{h}{r} F\left(\frac{\beta + \psi}{2} \,\middle|\, \frac{-4(1 - \frac{h}{r})}{(\frac{h}{r})^2}\right)$$

$$+ \frac{2\left|\frac{h}{r}\right|(1 - \frac{h}{r})}{2 - \frac{h}{r}} \frac{\sin(\beta + \psi)}{\sqrt{1 + (1 - \frac{h}{r})^2 - 2(1 - \frac{h}{r})\cos(\beta + \psi)}} + C \tag{B8}$$





$$\overline{G}_{II,ip}(\frac{h}{r}, \psi, \beta) = - \frac{(\frac{h}{r})^2}{(1-\frac{h}{r})(2-\frac{h}{r})} E\left(\frac{\beta+\psi}{2} \,\Big|\, \frac{-4(1-\frac{h}{r})}{(\frac{h}{r})^2}\right) - \frac{\frac{h}{r}}{1-\frac{h}{r}} F\left(\frac{\beta+\psi}{2} \,\Big|\, \frac{-4(1-\frac{h}{r})}{(\frac{h}{r})^2}\right)$$

$$+ \frac{\frac{h|h|}{r^2}\left(\frac{\beta}{1-\frac{h}{r}} - 2\frac{\sin(\beta+\psi)}{(2-\frac{h}{r})\frac{h}{r}}\right)}{\sqrt{1+(1-\frac{h}{r})^2 - 2(1-\frac{h}{r})\cos(\beta+\psi)}} + C \tag{B9}$$

In Eqs. (B8) and (B9), $F(x \mid m)$ and $E(x \mid m)$ are the incomplete elliptic integrals of the first and the second kind which are defined as follows:

$$F(x \mid m) = \int_0^x \frac{1}{\sqrt{1-m\sin^2(x)}}\, \mathrm{d}x \tag{B10}$$

$$E(x \mid m) = \int_0^x \sqrt{1-m\sin^2(x)}\, \mathrm{d}x \tag{B11}$$

The advantage of the derived analytical equations in the form of elliptic integrals over the original form is because of the existence of fast approximation methods, such as the work by Bulirsch (1965) and Fukushima (2012). With these computation efficient estimations, results with high accuracy can be obtained with a small fraction of the computational cost compared to

using direct numerical integration with Euler method or Runge–Kutta methods.

The analytical steady-state results for the special condition of in-plane trailed vorticity can then be calculated with low computational efforts. The normalized steady-state value of the base near-wake induction is:

$$\overline{W}_{I,ip} = \overline{G}_{I,ip}\left(\frac{h}{r}, \psi, \frac{\pi}{2}\right) - \overline{G}_{I,ip}\left(\frac{h}{r}, \psi, 0\right) \tag{B12}$$

$$\overline{W}_{II,ip} = \overline{G}_{II,ip}\left(\frac{h}{r}, \psi, \frac{\pi}{2}\right) - \overline{G}_{II,ip}\left(\frac{h}{r}, \psi, 0\right) \tag{B13}$$

To be noted, for this special condition of in-plane trailed vorticity, the near-wake which is the first quarter revolution of the wake of the own blade, is equivalent to one-quarter of a vortex ring.

The reason of defining the first quarter revolution as near-wake possibly origins from the introducing of the near-wake model by Beddoes (1987), which was for the application of helicopter aerodynamics. The ordinary helicopters are equipped with four blades, and one blade will encounter the wake of the previous blade with about 90° of azimuthal angle. The definition of

the near-wake part can be adjusted to other values. For the ordinary wind turbines which are equipped with three blades, the reader may argue that the definition could then be adjusted to 120°. If so, the steady-state value of the newly defined near-wake induction could be calculated using Eqs. (B12) and (B13) with the integral calculated until $\frac{2}{3}\pi$. And of course, the influence coefficient tensors for the convective correction in Sect. 5 need to be updated accordingly. However, it is not possible to argue that using the value of 120° is more physical comparing to using the value of 90° as in the current implementation. The

definition of the near-wake should not be connected to the number of blades. Instead, it is only an arbitrary split of the vortex wake domain and should ensure the near-wake part contains the near-wake effects. For example, the changed trailed vorticity





starting position due to blade sweep should be in the near-wake part. In a test that is not reported in this work, the results from the coupled method with either definition of the near-wake are very similar.

### B1.1 Relationship between inductions

Comparing the steady-state value of the axial and tangential near-wake base induction in Eqs. (B12) and (B13), the relationship
between them is as follows.

$$\overline{W}_{II,ip}\left(\frac{h}{r},\psi\right) = \frac{1}{1-\frac{h}{r}}\left(-\overline{W}_{I,ip}\left(\frac{h}{r},\psi\right) + \frac{h|h|}{r^2}\frac{\pi}{2\sqrt{1+(1-\frac{h}{r})^2-2(1-\frac{h}{r})\sin\psi}}\right) \tag{B14}$$

It has been proposed by Pirrung et al. (2016) to use the same value of $\Phi$ for the axial and tangential induction. With the new definition of $\Phi$ explained in Sect. 4, it is equivalent to assume $W_{I,ip}$ and $-W_{II,ip}$ are equal (the negative sign is inherited from the definition of the coordinate system). According to Pirrung et al. (2016), this assumption introduces only a small error for
straight blades when $\left|\frac{h}{r}\right|$ is small but will gradually deviate from the analytical results with the increase of $\left|\frac{h}{r}\right|$. This conclusion can also be obtained analytically according to Eq. (B14). For the straight blade, the value of $\psi$ is zero, Eq. (B14) is simplified as follows:

$$\overline{W}_{II,ip}\left(\frac{h}{r},\psi=0\right) = \frac{1}{1-\frac{h}{r}}\left(-\overline{W}_{I,ip}\left(\frac{h}{r},\psi=0\right) + \frac{\pi}{2}\frac{h}{r}\right) \tag{B15}$$

According to Eq. (B15), when the value of $\left|\frac{h}{r}\right|$ is small, $W_{I,ip}$ is approximately equal to $-W_{II,ip}$.

## B2  Straight trailed vorticity

For the special condition of straight trailed vorticity ($\varphi = \frac{\pi}{2}$), the base trailing function could be expressed using the relationship of $ds = r\,d\beta^*$. To be noted, now $d\beta = d\beta\cos\varphi = 0$.

$$dw_{I,ss} = \frac{\Delta\Gamma r}{4\pi h|h|}\frac{h|h|}{r^2}\frac{1-(1-\frac{h}{r})\cos\psi}{\left[1+(1-\frac{h}{r})^2-2(1-\frac{h}{r})\cos\psi+\beta^{*2}\right]^{\frac{3}{2}}}\,d\beta^* \tag{B16}$$

$$dw_{II,ss} = \frac{\Delta\Gamma r}{4\pi h|h|}\frac{h|h|}{r^2}\frac{1-\frac{h}{r}-\cos\psi}{\left[1+(1-\frac{h}{r})^2-2(1-\frac{h}{r})\cos\psi+\beta^{*2}\right]^{\frac{3}{2}}}\,d\beta^* \tag{B17}$$

Integrating the base trailing function in Eqs. (B16) and (B17) with $\beta$ from 0 to $\frac{\pi}{2}$ is equivalent to integrating with $\beta^*$ from 0 to infinity.

$$W_{I,ss} = \int_{\beta=0}^{\beta=\frac{\pi}{2}} dw_{I,ss} = \frac{\Delta\Gamma r}{4\pi h|h|}\int_0^\infty \frac{h|h|}{r^2}\frac{1-(1-\frac{h}{r})\cos\psi}{\left[1+(1-\frac{h}{r})^2-2(1-\frac{h}{r})\cos\psi+\beta^{*2}\right]^{\frac{3}{2}}}\,d\beta^* \tag{B18}$$

$$W_{II,ss} = \int_{\beta=0}^{\beta=\frac{\pi}{2}} dw_{II,ss} = \frac{\Delta\Gamma r}{4\pi h|h|}\int_0^\infty \frac{h|h|}{r^2}\frac{1-\frac{h}{r}-\cos\psi}{\left[1+(1-\frac{h}{r})^2-2(1-\frac{h}{r})\cos\psi+\beta^{*2}\right]^{\frac{3}{2}}}\,d\beta^* \tag{B19}$$





The definite integrals are derived as follows. They correspond to the base induction of a semi-infinite line vortex.

$$W_{I,ss} = \frac{\Delta\Gamma r}{4\pi h|h|} \frac{h|h|}{r^2} \frac{1 - (1 - \frac{h}{r})\cos\psi}{1 + (1 - \frac{h}{r})^2 - 2(1 - \frac{h}{r})\cos\psi} \tag{B20}$$

$$W_{II,ss} = \frac{\Delta\Gamma r}{4\pi h|h|} \frac{h|h|}{r^2} \frac{1 - \frac{h}{r} - \cos\psi}{1 + (1 - \frac{h}{r})^2 - 2(1 - \frac{h}{r})\cos\psi} \tag{B21}$$

So, the normalized based axial and tangential induction for this special condition of straight trailed vorticity are:

$$5 \quad \Phi_{I,ss} = \frac{1}{\frac{A_1}{b_1} + \frac{A_2}{b_2}} \frac{h|h|}{r^2} \frac{1 - (1 - \frac{h}{r})\cos\psi}{1 + (1 - \frac{h}{r})^2 - 2(1 - \frac{h}{r})\cos\psi} \tag{B22}$$

$$\Phi_{II,ss} = \frac{-1}{\frac{A_1}{b_1} + \frac{A_2}{b_2}} \frac{h|h|}{r^2} \frac{1 - \frac{h}{r} - \cos\psi}{1 + (1 - \frac{h}{r})^2 - 2(1 - \frac{h}{r})\cos\psi} \tag{B23}$$

The derived analytical equations are further analyzed. Firstly, for the condition of $\varphi = \frac{\pi}{2}$, the steady-state values of the axial and tangential induction will have the following value:

$$W_{x,ss} = W_{I,ss}\cos\varphi = 0 \tag{B24}$$

$$10 \quad W_{y,ss} = W_{II,ss}\sin\varphi = W_{II,ss} \tag{B25}$$

Secondly, the relationship between the normalized base induction of $\Phi_{I,ss}$ and $\Phi_{II,ss}$ are derived as follows.

$$\frac{\Phi_{I,ss}}{\Phi_{II,ss}} = -\frac{1 - (1 - \frac{h}{r})\cos\psi}{1 - \frac{h}{r} - \cos\psi} \tag{B26}$$

For the special condition that the blade is straight without sweep, which means $\psi = 0$, the two normalized base inductions are equal. This corresponds to using the same base axial and tangential induction of $\Phi_s$ for the straight blade as in the previous
15  work of Pirrung et al. (2017b).





## Appendix C: Results of the distributed load

### C1    backward swept blades

The difference of the loads of the backward swept blades (Blade-2 to Blade-4) compared to the baseline straight blade.

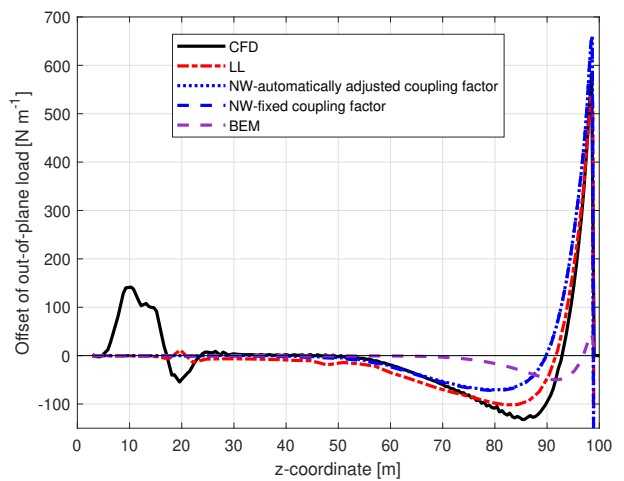
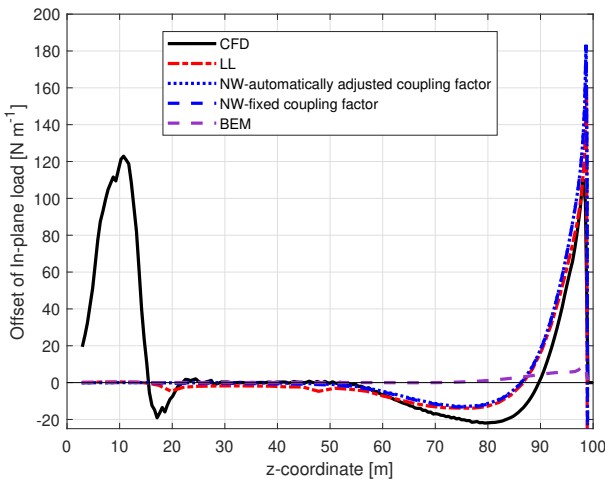

**Figure C1.** Comparison of the difference between the out-of-plane load (left) and the in-plane load (right) of the backward swept Blade-2 with respect to the baseline blade calculated from the Navier-Stokes solver (CFD), the lifting-line method (LL), the proposed coupled method (NW) and the blade element momentum method (BEM).

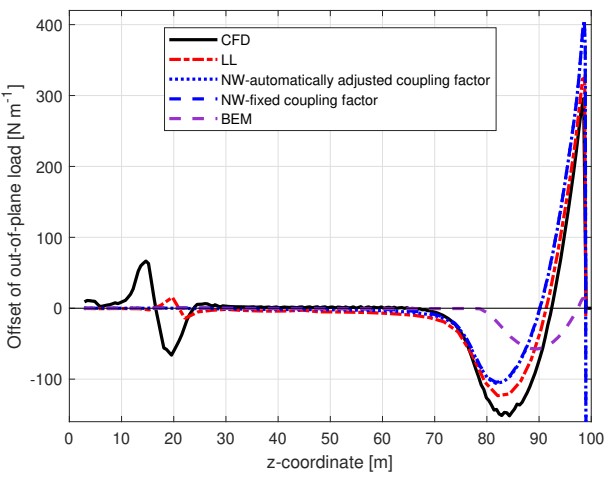
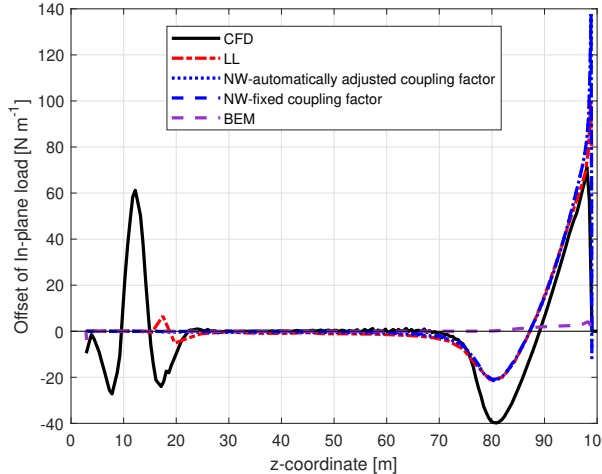

**Figure C2.** Comparison of the difference between the out-of-plane load (left) and the in-plane load (right) of the backward swept Blade-3 with respect to the baseline blade calculated from the Navier-Stokes solver (CFD), the lifting-line method (LL), the proposed coupled method (NW) and the blade element momentum method (BEM).





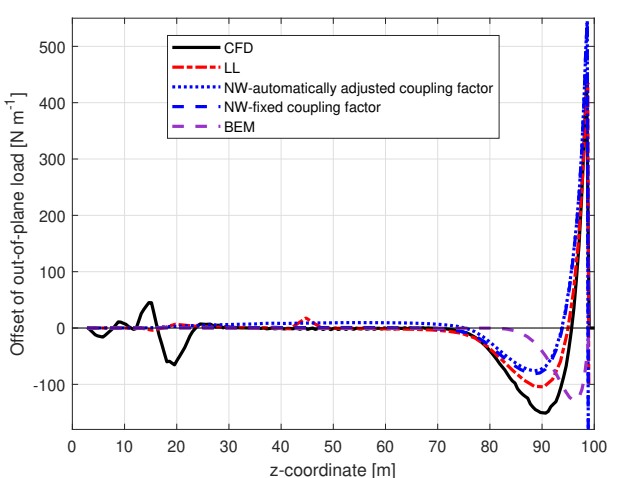
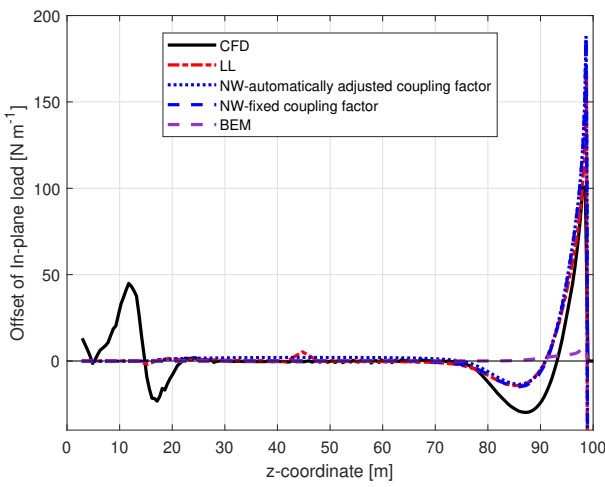

**Figure C3.** Comparison of the difference between the out-of-plane load (left) and the in-plane load (right) of the backward swept Blade-4 with respect to the baseline blade calculated from the Navier-Stokes solver (CFD), the lifting-line method (LL), the proposed coupled method (NW) and the blade element momentum method (BEM).



## C2 Forward swept blades

The difference of the loads of the forward swept blades (Blade-6 to Blade-8) compared to the baseline straight blade.

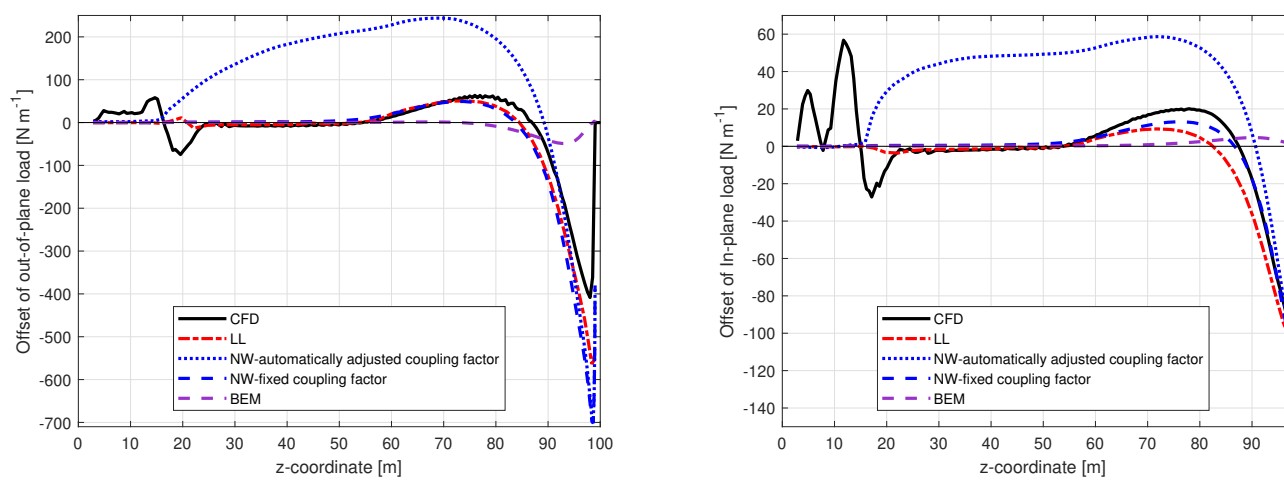

**Figure C4.** Comparison of the difference between the out-of-plane load (left) and the in-plane load (right) of the forward swept Blade-6 with respect to the baseline blade calculated from the Navier-Stokes solver (CFD), the lifting-line method (LL), the proposed coupled method (NW) and the blade element momentum method (BEM).

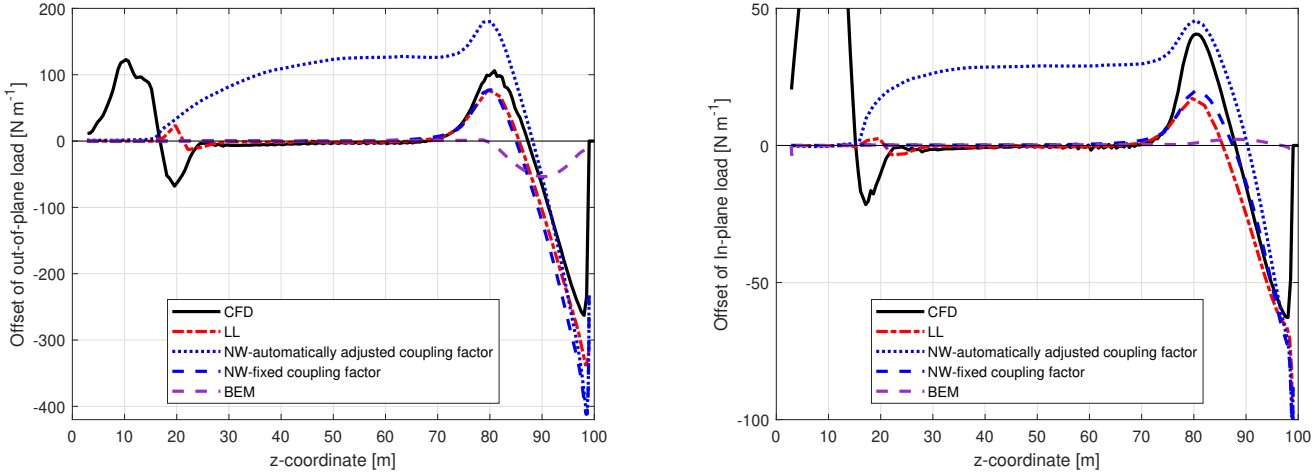

**Figure C5.** Comparison of the difference between the out-of-plane load (left) and the in-plane load (right) of the forward swept Blade-6 with respect to the baseline blade calculated from the Navier-Stokes solver (CFD), the lifting-line method (LL), the proposed coupled method (NW) and the blade element momentum method (BEM).





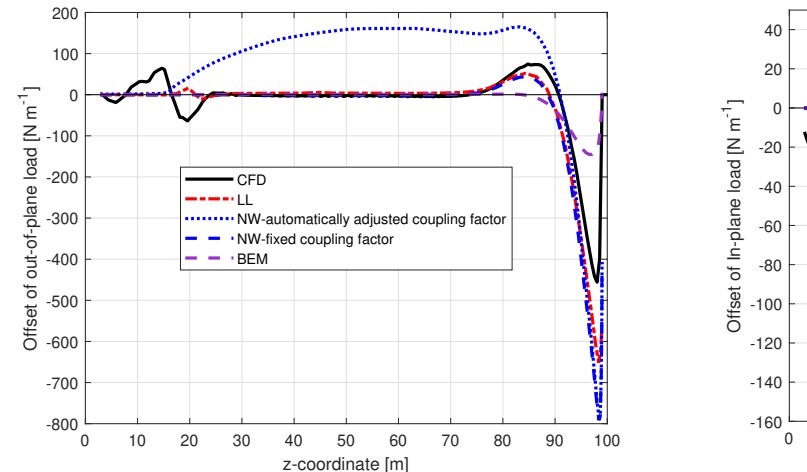 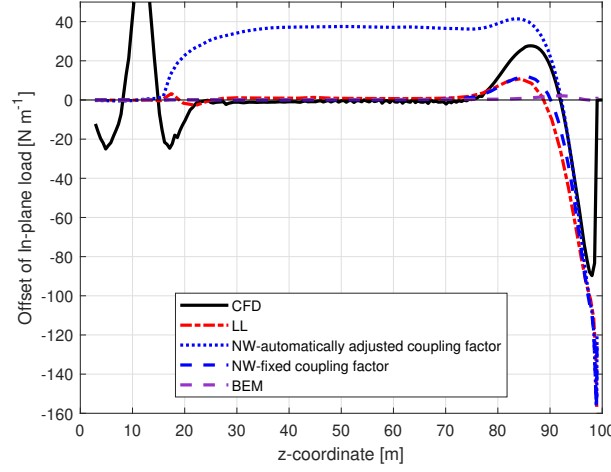

**Figure C6.** Comparison of the difference between the out-of-plane load (left) and the in-plane load (right) of the forward swept Blade-8 with respect to the baseline blade calculated from the Navier-Stokes solver (CFD), the lifting-line method (LL), the proposed coupled method (NW) and the blade element momentum method (BEM).

## Appendix D: Influence coefficient tensor

The influence coefficient tensor in double-precision floating-point format with full digits can be found in the online supplement (Li et al., 2021). The coefficients shown here are rounded to six to eight decimals with slightly reduced accuracy. The relative error of the convective correction with the full digits and the reduced digits using the following coefficients are summarized in
5    the following table.

**Table D1.** The relative error of the convective correction using the influence coefficients with full digits or reduced digits.

| Name | Influence coefficient | Maximum error using full digits | Maximum error with reduced digits |
|---|---|---|---|
| Region a1 | $\mathbf{I^{a1}}$ | 0.78% | 0.78% |
| Region a2 | $\mathbf{I^{a2}}$ | 1.10% | 1.10% |
| Region a3 | $\mathbf{I^{a3}}$ | 1.34% | 1.43% |
| Region t1 | $\mathbf{I^{t1}}$ | 0.54% | 0.54% |
| Region t2 | $\mathbf{I^{t2}}$ | 0.95% | 0.96% |



## D1 Influence coefficient tensors for axial induction

$$\mathbf{I^{a1}}(1,:,:) = \begin{pmatrix} -30.2629953 & -44.0123379 & -4.8091034 & 6.6816625 & -1.8728117 \\ 209.4691684 & 342.2487285 & 96.4620698 & -5.7254494 & 13.6705749 \\ -290.1750741 & -466.0937921 & -181.7678232 & -53.0768773 & -33.3088719 \\ 83.8736848 & 110.3440879 & 43.9267968 & 26.7977770 & 16.8236210 \\ 172.3498017 & 197.0336475 & 40.8145840 & 2.7259789 & 7.9637918 \\ -287.1461269 & -409.7263812 & -148.6834315 & -57.3376482 & -33.2609586 \\ 84.2660443 & 106.2805603 & 41.6369816 & 27.4446775 & 16.7602353 \end{pmatrix} \tag{D1}$$

$$\mathbf{I^{a1}}(2,:,:) = \begin{pmatrix} 53.1550561 & 82.7950371 & 11.8470979 & -2.5812529 & 12.4529125 \\ -418.9619277 & -741.0768866 & -285.1483971 & -79.7753292 & -72.2472370 \\ 587.9307272 & 1017.3608691 & 491.3189773 & 241.8729167 & 142.5305922 \\ -137.1825739 & -200.3207669 & -107.8391565 & -95.5704421 & -70.4028947 \\ -375.7864254 & -416.3134514 & -191.2711984 & -97.3931617 & -40.4700946 \\ 603.2260980 & 873.7974991 & 406.5491994 & 252.9426783 & 135.4379812 \\ -140.6835481 & -188.9546718 & -101.1232811 & -98.1065552 & -69.6831846 \end{pmatrix} \tag{D2}$$

$$\mathbf{I^{a1}}(3,:,:) = \begin{pmatrix} -24.4634404 & -49.1773459 & -15.2835165 & -18.7602840 & -20.2094150 \\ 257.8948363 & 528.3100345 & 298.1358374 & 186.0849450 & 109.0304987 \\ -371.0131449 & -740.4358281 & -483.9445448 & -364.1513924 & -200.5487163 \\ 45.2643636 & 72.7402636 & 72.2423835 & 128.0351116 & 106.8599904 \\ 267.5458324 & 277.6881446 & 258.8159075 & 168.7618522 & 57.0927392 \\ -408.5386036 & -612.6519838 & -409.0961438 & -368.8174390 & -185.3398947 \\ 51.0719807 & 61.6212110 & 65.5878879 & 130.9154917 & 105.4798178 \end{pmatrix} \tag{D3}$$

$$\mathbf{I^{a1}}(4,:,:) = \begin{pmatrix} 0.9777967 & 8.7243693 & 9.2512766 & 19.2782525 & 11.0076639 \\ -47.3025925 & -129.3430362 & -116.4275187 & -117.7402703 & -57.1884276 \\ 73.7522569 & 195.6121511 & 195.7705625 & 212.9970038 & 108.3347082 \\ 9.3193829 & 8.5832630 & -23.5309335 & -81.8175506 & -71.0052089 \\ -67.4079250 & -56.4117165 & -114.5683550 & -71.9868045 & -25.8741364 \\ 98.6913683 & 149.3325565 & 169.7441677 & 210.0891275 & 97.6306099 \\ 5.8410702 & 13.0063375 & -20.9015181 & -82.9690385 & -70.1169804 \end{pmatrix} \tag{D4}$$

$$\mathbf{I^{a1}}(5,:,:) = \begin{pmatrix} 0.3905372 & 0.0863371 & 0.0809883 & -1.8632991 & -0.3010930 \\ 0.5059303 & 5.3663399 & 6.7255071 & 13.3051729 & 5.0750717 \\ -2.5363161 & -14.3742774 & -19.8092364 & -30.4294738 & -13.6991207 \\ -0.1478574 & 8.1063946 & 19.9638191 & 31.0576061 & 21.0751565 \\ 4.9304429 & -1.0082621 & 10.6365516 & 3.2151968 & 3.0734356 \\ -8.4771690 & -7.3161932 & -17.9411512 & -29.4501544 & -10.7083398 \\ 0.6641002 & 7.4457629 & 19.5451515 & 31.2496327 & 20.9271491 \end{pmatrix} \tag{D5}$$

$$\mathbf{I^{a1}}(6,:,:) = \begin{pmatrix} -0.0507499 & 0.0558777 & 0.0317971 & -0.0089532 & 0.0002913 \\ 0.1286709 & -0.3151871 & 0.0018811 & 0.0985785 & -0.0865264 \\ -0.0172141 & 0.6809711 & -0.1988941 & -0.2182092 & 0.2773425 \\ -0.1313101 & -0.5705022 & 0.2361800 & 0.2094688 & -0.6421827 \\ -0.1675725 & 0.2054625 & -0.3660916 & 0.3618960 & -0.0859911 \\ 0.4346966 & -0.1987679 & 0.6714691 & -0.8470430 & 0.2767341 \\ -0.1938734 & -0.5281251 & 0.2743414 & 0.1859921 & -0.6412762 \end{pmatrix} \tag{D6}$$

$$\mathbf{I^{a2}}(1,:,:) = \begin{pmatrix} 14.1443428 & -3.4200336 & -8.6632766 & -3.2512917 & 1.6673927 \\ -14.3379998 & -42.4526269 & 44.8844649 & 1.8129803 & 2.9789316 \\ -34.0068718 & 143.5109416 & -83.6124141 & 6.9214482 & -16.4443492 \\ 12.7301783 & -60.5729910 & 23.6875815 & 3.0630625 & 10.5710667 \\ -11.7085861 & -20.3817370 & 10.6563746 & 15.1727425 & 4.6042797 \\ 9.3462571 & 69.2676499 & -32.9332761 & -20.4817802 & -14.2982153 \\ 9.3111198 & -53.6979766 & 17.2284326 & 8.9936123 & 9.3346192 \end{pmatrix} \tag{D7}$$





$$\mathbf{I^{a2}}(2,:,:) = \begin{pmatrix} -39.3288761 & 17.7574352 & 18.7182283 & 10.1844694 & -8.5144460 \\ 47.6559011 & 80.3132512 & -106.3400261 & -10.1038357 & 6.3197326 \\ 72.4404153 & -333.6400291 & 221.1504756 & -30.6689736 & 27.4295624 \\ -21.8083886 & 117.2183132 & -47.3949105 & 0.7193395 & -19.6817720 \\ 54.9226623 & 26.9319933 & -35.3166109 & -35.3811073 & -8.2273724 \\ -68.9432298 & -109.4456982 & 85.0824950 & 35.4281485 & 26.4170866 \\ -7.2598922 & 91.9682848 & -28.2137310 & -13.8716215 & -17.0665052 \end{pmatrix} \tag{D8}$$

$$\mathbf{I^{a2}}(3,:,:) = \begin{pmatrix} 39.2147049 & -29.0669181 & -9.4187881 & -11.5897887 & 13.0003672 \\ -54.9123080 & -30.4177512 & 74.2365320 & 18.2797806 & -25.9860317 \\ -51.2327615 & 243.2741899 & -184.7819962 & 34.7735741 & 0.3193281 \\ 12.0319589 & -53.5018994 & 12.2342678 & -5.4489537 & 6.8592144 \\ -83.1932889 & 19.0080018 & 32.4884520 & 33.5003977 & -0.2699424 \\ 118.5264133 & -9.2291737 & -51.9070516 & -23.4928839 & -3.5990547 \\ -8.9078459 & -20.0167245 & -9.2091721 & 7.7736824 & 4.9798854 \end{pmatrix} \tag{D9}$$

$$\mathbf{I^{a2}}(4,:,:) = \begin{pmatrix} -16.6247282 & 17.1322393 & 0.6646976 & 3.7853652 & -7.1490439 \\ 26.6945298 & -9.4634984 & -20.9011874 & -5.9400662 & 20.8141528 \\ 11.6620329 & -62.4467128 & 72.6537901 & -27.6371035 & -14.4197743 \\ -2.5014749 & 1.9771639 & -5.1332573 & 17.6369316 & -0.9305895 \\ 52.4970277 & -35.2171443 & -16.6820337 & -12.2574820 & 5.4664956 \\ -79.6740851 & 64.2142021 & 22.5005971 & -5.9624868 & -9.9836695 \\ 10.2663768 & -16.9965935 & 4.9866977 & 12.5326197 & -0.3982437 \end{pmatrix} \tag{D10}$$

$$\mathbf{I^{a2}}(5,:,:) = \begin{pmatrix} 2.1680909 & -1.9325002 & -1.4598676 & 0.9013355 & 1.0045864 \\ -3.6267041 & 1.3746512 & 7.1708105 & -3.3173219 & -4.3791033 \\ -1.4309188 & 8.1720898 & -20.9017965 & 14.7674750 & 3.2422534 \\ 1.6288906 & -3.7561081 & 13.0588432 & -15.3645870 & 3.5123401 \\ -12.2546955 & 11.8229409 & 6.7489696 & 0.7852945 & -1.5344478 \\ 19.8474127 & -19.7529300 & -16.9628675 & 10.8830814 & 1.1630907 \\ -1.5910305 & 0.6576136 & 11.2469182 & -14.6630860 & 3.4684830 \end{pmatrix} \tag{D11}$$

10 $$\mathbf{I^{a2}}(6,:,:) = \begin{pmatrix} 0.2486922 & -0.8554398 & 1.0648231 & -0.4566182 & -0.0001123 \\ -0.7408567 & 2.4571284 & -3.0863117 & 1.1621515 & 0.1275147 \\ 1.1255059 & -3.7190413 & 5.3021715 & -2.7304257 & 0.1847216 \\ -0.9427155 & 3.2675639 & -5.3772642 & 3.5408975 & -1.1829603 \\ 0.2071953 & -0.2041765 & -1.6546668 & -0.0699594 & 0.1263416 \\ -0.5694562 & -1.2096613 & 6.0023983 & -1.8571029 & 0.1871016 \\ -0.6754359 & 2.9189806 & -5.2705428 & 3.5298212 & -1.1843152 \end{pmatrix} \tag{D12}$$

$$\mathbf{I^{a3}}(1,:,:) = \begin{pmatrix} 9.11665188 & 29.79222047 & 43.86616973 & 32.40470861 & 11.09564270 \\ -26.09418131 & -155.76996750 & -205.31734409 & -95.53860789 & -49.42888125 \\ 17.18011985 & 321.09112364 & 440.56740600 & 134.73797533 & 79.23027899 \\ 1.78046701 & -235.24443181 & -354.35998617 & -100.81604882 & -45.41461529 \\ -27.27974290 & -160.15698589 & -194.90945090 & -60.95849006 & -26.56400855 \\ 55.19661745 & 429.90429486 & 542.61100878 & 160.40424204 & 77.17323077 \\ -2.99560189 & -249.51706273 & -368.53902302 & -104.58605057 & -44.83785553 \end{pmatrix} \tag{D13}$$

$$\mathbf{I^{a3}}(2,:,:) = \begin{pmatrix} -28.19478766 & -88.29006039 & -135.04025383 & -112.21215802 & -44.11700442 \\ 66.96477017 & 379.25990135 & 508.07689753 & 283.97781693 & 176.57597505 \\ 17.66091431 & -634.74312353 & -945.13710121 & -312.12785403 & -248.40124366 \\ -82.78952352 & 402.54894986 & 750.61182255 & 222.12300396 & 125.19654785 \\ 71.21588008 & 400.15306970 & 479.60405330 & 159.55305390 & 88.70259352 \\ -107.50498855 & -986.13956564 & -1267.70442454 & -391.90258276 & -243.62046718 \\ -66.59092597 & 449.80615250 & 796.55179975 & 234.22262799 & 123.53525679 \end{pmatrix} \tag{D14}$$





$$\mathbf{I^{a3}}(3,:,:) = \begin{pmatrix} 32.04325775 & 94.31136564 & 151.89716069 & 145.88015244 & 65.69149913 \\ -41.92639331 & -289.17519494 & -453.44136049 & -340.49327439 & -229.24935079 \\ -170.30486123 & 215.16871649 & 648.01599624 & 302.01030822 & 259.55461252 \\ 245.82143715 & 11.51095552 & -482.77494741 & -196.51499653 & -96.26782702 \\ -48.71461830 & -325.02218908 & -426.10623881 & -172.15231906 & -102.70304777 \\ -13.09916989 & 644.38829988 & 1029.98579041 & 395.51384357 & 257.36267246 \\ 224.39180264 & -48.81548796 & -539.49667228 & -211.33054661 & -94.66534389 \end{pmatrix}$$ (D15)

$$\mathbf{I^{a3}}(4,:,:) = \begin{pmatrix} -16.73843958 & -44.56433162 & -75.42026032 & -85.39941222 & -43.76287003 \\ -14.70745503 & 44.77891622 & 172.30670573 & 200.56405072 & 128.25861363 \\ 233.25304850 & 271.12617385 & -101.20880449 & -163.39550356 & -94.03984994 \\ -264.95000244 & -365.04228466 & 22.89248374 & 90.84475815 & 0.62328159 \\ -9.84311242 & 69.08022171 & 158.74599828 & 97.55666592 & 47.05250048 \\ 140.04161225 & 26.26812272 & -309.83694548 & -214.97805164 & -95.91885365 \\ -251.04680957 & -327.70311499 & 56.48247681 & 99.70072067 & 0.14671554 \end{pmatrix}$$ (D16)

$$\mathbf{I^{a3}}(5,:,:) = \begin{pmatrix} 4.50261103 & 10.40079158 & 16.02776422 & 19.92248023 & 11.41551847 \\ 19.08956817 & 27.19876620 & -18.83043604 & -48.79174767 & -26.44677059 \\ -115.15085794 & -212.28073597 & -71.37815649 & 31.05294396 & 2.57760514 \\ 116.42073298 & 226.69126139 & 97.46889829 & -5.07507500 & 17.02834139 \\ 18.31219088 & 23.89526985 & -12.19577177 & -22.95750203 & -6.20474382 \\ -89.27812582 & -147.60473128 & -19.25411563 & 44.26364549 & 3.53714562 \\ 111.92063809 & 215.18207328 & 87.51679466 & -7.88527165 & 16.94297940 \end{pmatrix}$$ (D17)

$$\mathbf{I^{a3}}(6,:,:) = \begin{pmatrix} -0.73071849 & -1.65270094 & -1.33216874 & -0.59618790 & -0.32276422 \\ -3.32244087 & -6.71096091 & -3.48162012 & 0.37331021 & 0.62710277 \\ 17.37007235 & 41.01276696 & 31.35433242 & 7.45124387 & 0.42793410 \\ -16.30032357 & -41.58074598 & -35.61364526 & -10.36033165 & -1.61111293 \\ -3.69169305 & -8.37718404 & -5.83095458 & -0.95126151 & 0.05345689 \\ 14.65722437 & 34.56697920 & 26.41672645 & 6.42924353 & 0.81554890 \\ -15.69702678 & -40.06982556 & -34.29089544 & -9.92025766 & -1.56710772 \end{pmatrix}$$ (D18)

## D2 Influence coefficient tensors for tangential induction

$$\mathbf{I^{t1}}(1,:,:) = \begin{pmatrix} -10.950183 & -8.461191 & 4.898831 & -0.274851 & -2.163785 \\ 73.495139 & 91.960394 & 21.537951 & 17.291370 & 12.936439 \\ -110.228670 & -151.928768 & -61.034637 & -54.434259 & -25.629843 \\ 42.002623 & 53.451397 & 19.189837 & 25.672729 & 13.793905 \\ 56.614278 & 56.724787 & 19.428838 & 17.362819 & 8.835412 \\ -114.732203 & -131.581665 & -53.717208 & -57.978611 & -26.616543 \\ 43.141667 & 51.648530 & 19.382676 & 26.443906 & 14.043129 \end{pmatrix}$$ (D19)

$$\mathbf{I^{t1}}(2,:,:) = \begin{pmatrix} 23.949901 & 14.778852 & -11.888094 & 2.752459 & 8.202003 \\ -172.382067 & -223.146941 & -82.226496 & -64.009011 & -46.870889 \\ 261.465264 & 389.567442 & 216.370674 & 174.558054 & 90.985548 \\ -90.416434 & -132.858435 & -82.922090 & -88.970970 & -54.008049 \\ -137.165438 & -142.967414 & -89.290104 & -69.621141 & -28.857329 \\ 277.999420 & 331.402610 & 192.070864 & 189.634603 & 90.450645 \\ -94.163160 & -127.374361 & -81.733697 & -92.222698 & -54.397761 \end{pmatrix}$$ (D20)

$$\mathbf{I^{t1}}(3,:,:) = \begin{pmatrix} -16.458425 & -5.233017 & 6.562594 & -10.476138 & -11.684835 \\ 133.610420 & 180.884019 & 114.983093 & 103.098665 & 64.356335 \\ -207.804511 & -344.267998 & -282.408473 & -238.886879 & -123.527042 \\ 60.831235 & 109.376515 & 119.761288 & 131.154016 & 81.095594 \\ 112.486039 & 123.270583 & 129.753159 & 91.206345 & 34.958736 \\ -227.936935 & -288.904927 & -256.603938 & -251.524286 & -117.483340 \\ 65.084160 & 103.807907 & 117.344461 & 134.914057 & 80.907908 \end{pmatrix}$$ (D21)





$$\mathbf{I^{t1}}(4,:,:) = \begin{pmatrix} 3.712240 & -1.698233 & 0.689174 & 9.357443 & 5.915291 \\ -37.247274 & -53.083017 & -63.043242 & -66.669945 & -34.242303 \\ 61.592618 & 120.349330 & 152.800477 & 144.112769 & 68.456084 \\ -13.907175 & -39.307187 & -77.574138 & -93.356087 & -55.044777 \\ -35.216502 & -39.157094 & -69.722510 & -43.639326 & -17.534227 \\ 72.268870 & 99.566991 & 142.017545 & 145.864151 & 62.263219 \\ -16.012744 & -37.054272 & -76.259003 & -94.838522 & -54.602871 \end{pmatrix} \tag{D22}$$

$$\mathbf{I^{t1}}(5,:,:) = \begin{pmatrix} -0.255738 & 0.571123 & -0.194635 & -1.438571 & -0.295277 \\ 2.643824 & 3.613584 & 8.299555 & 10.505772 & 3.820332 \\ -5.306676 & -13.998280 & -24.701326 & -25.716357 & -10.470962 \\ 1.663405 & 9.461890 & 20.835279 & 25.597187 & 14.453228 \\ 3.583366 & 1.928772 & 10.154802 & 4.368168 & 1.977129 \\ -8.286969 & -10.041223 & -23.923139 & -25.670827 & -7.835865 \\ 2.184772 & 9.070745 & 20.557907 & 25.805052 & 14.333351 \end{pmatrix} \tag{D23}$$

$$\mathbf{I^{t1}}(6,:,:) = \begin{pmatrix} -0.055049 & 0.035890 & 0.032544 & -0.000811 & 0.000342 \\ 0.145594 & -0.246526 & -0.007581 & 0.068080 & -0.091463 \\ -0.068985 & 0.542979 & -0.215523 & -0.162468 & 0.306337 \\ -0.080029 & -0.460504 & 0.256993 & 0.150341 & -0.705551 \\ -0.171522 & 0.270405 & -0.379745 & 0.352277 & -0.090620 \\ 0.400601 & -0.349407 & 0.632882 & -0.772935 & 0.305331 \\ -0.148020 & -0.414416 & 0.299683 & 0.122614 & -0.704614 \end{pmatrix} \tag{D24}$$

$$\mathbf{I^{t2}}(1,:,:) = \begin{pmatrix} -5.7258055 & 4.6742091 & 3.9009208 & -3.6012671 & 1.6494929 \\ 22.2232546 & -36.6333746 & 5.7078416 & 6.8000797 & -8.0609420 \\ -12.3848827 & 67.0809077 & -58.2052203 & 6.0215690 & 18.9447639 \\ -13.8146709 & -28.7098857 & 63.3541108 & -15.2928387 & -14.8005194 \\ -1.9457799 & -21.0008695 & 36.6230667 & -6.9262636 & -5.9263411 \\ 12.2705756 & 50.2206921 & -106.6422066 & 23.5785134 & 20.4626135 \\ -18.7356724 & -24.7651457 & 70.5192355 & -17.3834529 & -15.7073488 \end{pmatrix} \tag{D25}$$

$$\mathbf{I^{t2}}(2,:,:) = \begin{pmatrix} 10.3949253 & -1.7986434 & -13.5540153 & 6.0950126 & -5.2899093 \\ -50.4535447 & 68.6347658 & -5.9411855 & -1.4965935 & 23.5785440 \\ 29.8238964 & -130.9277484 & 124.4186812 & -47.2728909 & -48.6232937 \\ 39.5765002 & 32.8957644 & -136.0211033 & 66.2260029 & 31.8846485 \\ 0.6015098 & 42.7988005 & -89.0992084 & 31.0752719 & 15.4963826 \\ -25.5482064 & -82.3969757 & 250.9121459 & -97.9814698 & -50.0053027 \\ 50.8318678 & 22.3567994 & -154.2821458 & 72.4640619 & 34.1845044 \end{pmatrix} \tag{D26}$$

$$\mathbf{I^{t2}}(3,:,:) = \begin{pmatrix} -2.5395506 & -13.1033176 & 18.0894175 & -1.4842956 & 5.2035291 \\ 31.8741772 & -24.6952234 & -7.2920794 & -21.9824970 & -21.7548434 \\ -11.7055962 & 52.6048792 & -77.0772833 & 89.8101815 & 36.1605500 \\ -50.7802429 & 30.2485514 & 89.8583938 & -102.6158206 & -11.9793828 \\ 0.9231891 & -17.8464718 & 73.4366291 & -47.5554946 & -12.4124918 \\ 28.6677816 & -3.4839939 & -196.3401020 & 143.0821824 & 33.8401851 \\ -59.2778989 & 40.9278679 & 106.2721634 & -109.4699299 & -13.9627139 \end{pmatrix} \tag{D27}$$

$$\mathbf{I^{t2}}(4,:,:) = \begin{pmatrix} -3.5396780 & 13.7008953 & -10.3989969 & -2.2204264 & -1.4992357 \\ -1.7337036 & -12.3340251 & 8.6906327 & 25.7705289 & 5.6675673 \\ -9.9735983 & 20.0932861 & 10.2548742 & -69.4769647 & -2.4577836 \\ 30.6922787 & -43.8152447 & -20.2337737 & 73.5698952 & -12.7866025 \\ 0.3073784 & -8.1967548 & -23.8569770 & 31.5902276 & 2.1205812 \\ -18.4228389 & 52.2703527 & 60.1799648 & -93.0749226 & 0.9251449 \\ 32.7731899 & -48.8178908 & -26.2751530 & 76.9444402 & -12.1444514 \end{pmatrix} \tag{D28}$$



$$\mathbf{I^{t2}}(5,:,:) = \begin{pmatrix} 1.5827856 & -3.5265698 & 1.7437259 & 1.3796397 & -0.0692422 \\ -2.6055764 & 5.2706415 & -0.4417942 & -9.7260849 & 0.5830398 \\ 5.7375263 & -9.6398019 & -1.0522954 & 22.6003694 & -4.4651553 \\ -6.8976806 & 10.1399490 & 4.4808343 & -23.3338246 & 7.8224448 \\ -0.2557835 & 5.0101740 & 3.7705422 & -8.6121159 & 1.0503852 \\ 4.2707170 & -18.8232433 & -11.0601730 & 25.6977633 & -6.1702348 \\ -6.7868106 & 11.1837059 & 5.2291191 & -24.0665130 & 7.7724456 \end{pmatrix} \tag{D29}$$

$$\mathbf{I^{t2}}(6,:,:) = \begin{pmatrix} -0.0679784 & -0.0541334 & 0.1144018 & -0.1140784 & 0.0014493 \\ 0.1998278 & 0.2681714 & -0.2238743 & 0.3689901 & -0.0702087 \\ -0.2359921 & -0.5382265 & 0.3799151 & -0.9913595 & 0.5354574 \\ 0.0606999 & 0.4776227 & -0.2504437 & 0.8046238 & -0.7737097 \\ -0.1206340 & -0.2412165 & -0.3751788 & 0.1596029 & -0.0669662 \\ 0.2997150 & 0.5702797 & 1.3827055 & -0.4581764 & 0.5331981 \\ -0.0108322 & 0.4032303 & -0.2316729 & 0.8488675 & -0.7734641 \end{pmatrix} \tag{D30}$$

*Author contributions.* AL conducted the study as part of his PhD research. AL, GRP, MG and HAaM jointly developed the modified coupled near- and far- wake model. AL, GRP and MG contributed to the modification to the near-wake model. HAaM, GRP and MG contributed to the far-wake model and the coupling method. AL derived the analytical equations of steady-state near-wake induction in the form of elliptic integrals. AL proposed the method of generalized relative position and normalization of the sweep angle, with contributions from GRP and MG. The data fitting to calculate the influence coefficient tensors are performed by AL, with contributions from GRP. The results of the
coupled method and the BEM method are computed by AL. The lifting-line results are computed by AL. The CFD method is introduced by SGH and the CFD results are computed by SGH.

*Competing interests.* DTU Wind Energy develops and distributes HAWC2 on commercial and academic terms.

*Acknowledgements.* The authors would like to thank our colleague Néstor Ramos García for the help and suggestions in the lifting-line
simulation using the MIRAS code. The authors would also like to thank our colleague Alexander Meyer Forsting for discussions on the relevant topics.

This work has been supported by the Smart Tip project, funded by Innovation Fund Denmark (J.nr. 7046-00023B).



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
