# Peer review of "A computationally efficient engineering aerodynamic model for swept wind turbine blades"

_Wind Energy Science, 2021_

## Referee Comment (RC2)

[referee-annotated manuscript omitted]

---

## Author Comment (AC1)

November 17, 2021

Dear Reviewers

First of all we would like to thank you for the constructive and detailed comments on our article. Please find below our responses (in black) to your comments (in blue). In addition to changes suggested by the reviewers, we updated the labels of the y-axis in Sect. 8.5, 8.6 and Appendix C. The other content in the figures are not changed. Please find at the end of this letter a marked-up version showing all changes in the manuscript.

Yours sincerely

A. Li, G. R. Pirrung, M. Gaunaa, H. Aa. Madsen and S. G. Horcas

Technical University of Denmark            Frederiksborgvej 399            angl@dtu.dk
**Department of Wind Energy**            Building 125            www.dtu.dk
                                         DK-4000 Roskilde

[Figure]

**Comments to Reviewer #1**

The authors developed a new strategy for the simulation of Horizontal-Axis Wind Turbines (HAWTs) with swept blades, with the aim of reducing the computational cost associated to higher-order approaches such as Lifting Line Theory (LLT). The proposed method combines a near-wake model, based on the use of analytical solutions and approximations, with a far-wake one exploiting the Blade Element Momentum (BEM). An extensive testing was performed on a selected large size turbine for different blade degree of sweeping and the obtained predictions were compared with those of the BEM and two high-order methods, i.e. LLT and blade-resolved CFD, in terms of blade spanwise load distributions. A promising accuracy improvement is observed with the new method compared to traditional BEM.

The reviewer believes that the topic and the activity is very interesting, innovative and worthy of investigation. The adopted methodology is rigorous and clearly detailed throughout the whole paper, which is very well presented. Some specific considerations:

The authors claim that the developed method is computationally more efficient than the Lifting Line one, which is taken as a reference in the present work. It would be helpful for the reader to support this statement with some numerical data, for instance the computation time required by each method for the performed simulations

Based on the aforementioned comments, the publication of the paper in the present form is strongly recommended.

We included a new section (Sect. 8.8) that is dedicated to describe the computational effort of different aerodynamic models that are used in the present work. The high performance computing (HPC) cluster setup and the corresponding wall time of the RANS Navier-Stokes solver as well as the lifting-line method are described. The CPU time for the BEM method and the proposed coupled method used in the present work is also described. In addition, the CPU time of the proposed coupled method when used for an aeroelastic simulation and the comparison to the aeroelastic simulation using the BEM method are also described.

[Figure]

**Comments to Reviewer #2**

The paper presents a cost effective method for the modeling of swept rotor geometries, which can be used in routine aeroelastic certification simulations for wind turbines without penalizing overall computational cost. The model is based on previous developments by the authors, of a coupled near-far wake model, in which the near wake part modeling is based on a semi-analytical vortex filament representation while the far part is based on standard BEM. The authors provide a detailed description of the adaptations made to the original near wake model while they present improvements with respect to previous implementations.

They verify their model predictions by comparing load results of different swept blade shapes against a medium fidelity option (lifting line) and a high fidelity option (fully resolved CFD). They also convincingly demonstrate the improvement attained in the prediction of loads with respect to standard BEM implementations, which neglect wake induced effects due to wake filaments shifted position.

The paper is very well written. It is an original contribution, presenting a newly developed model, which can improve the fidelity of aeroelastic analyses. Along with some very few and minor comments which can be found in the accompanying pdf I would only recommend the authors to assess and present the overall effect on the predicted Cp by the new method in comparison to standard BEM. This is especially important given that, as well known and also highlighted in the results, standard BEM significantly underestimates the effect of the curved geometry on power output.

We included a new section (Sect. 8.7) that is dedicated to compare the overall effect on the prediction of aerodynamic power and thrust of backwards and forward swept blades by the BEM method, the lifting-line (LL) method and the proposed coupled method with fixed coupling factor. It can be concluded from the results that the proposed coupled method with fixed coupling factor is having improved agreement with the higher-fidelity LL method when predicting rotor power and thrust compared to the BEM method, for both backwards and forward swept blades.

Otherwise, the paper can be published as is.

There is one last comment but relevant to the bound vortex effect in LL methods which is not directly linked to the present model (therefore the authors are not asked to answer). I had to refer to the torque paper in order to understand the hybrid method you apply in order to calculate the bound vortex effect. In my opinion the application of a small core radius solves the problem and saves you from the complexity of the proposed idea. Our experience shows that in filament wake representations a very small core radius is needed in order to enhance the stability of the simulation. The same is good enough for filtering singularities due to the bound vortex.

Thank you for the comment on the bound vortex effect. We will look into your suggestion in the LL method in the near future.

Comments in the accompanying pdf:

**1) P5 Line 20: Along z axis, if negative then why do you need the -h1 in the plot? Again, why negative sign in the plot? It is a bit confusing.**

This is because in Fig. 2, the value of $h_1$ and $\psi_1$ are both negative. The values shown in the Fig. 2 are all positive values, so $-h_1$ and $-\psi_1$ are shown in the plot. We agree the text in the manuscript was not clear enough. Therefore, we added two sentences in Sect. 3 to better describe the definition of the signs of $h$ and $\psi$.

*The value of $h$ is positive when the calculation point is further inboard compared to the trailing point, and is negative when the calculation point is further outboard compared to the trailing point.*

*The value of $\psi$ is positive when the trailing point is azimuthally lagging behind the calculation point, and is negative when the trailing point is azimuthally leading ahead the calculation point.*

**2) P6 Line 2: Delete "which is equal to $r$ minus $r_{cp}$"**

Changed accordingly

**3) P6 Line 11: $\Omega r$ is also due to blade motion though!**

Thank you for the comment. We have updated the definition and the text to: *the velocities due to blade deformation* instead of: *the velocities due to blade motion*, in order to be more precise.

**4) P10 Line 21: Please re-phrase - use of English**

[revised manuscript text omitted]